

# Primary and secondary aerosols in Beijing in winter: sources, variations and processes

Yele Sun[1,2], Wei Du[1], Pingqing. Fu[1], Qingqing Wang[1], Jie Li[1], Xinlei Ge[3], Qi Zhang[4], Chunmao Zhu[5,6], Lujie Ren[1], Weiqi Xu[1], Jian Zhao[1], Tingting Han[1], D. R. Worsnop[7], Zifa Wang[1]

[1]State Key Laboratory of Atmospheric Boundary Layer Physics and Atmospheric Chemistry, Institute of Atmospheric Physics, Chinese Academy of Sciences, Beijing 100029, China

[2]Center for Excellence in Urban Atmospheric Environment, Institute of Urban Environment, Chinese Academy of Sciences, Xiamen 361021, China

[3]School of Environmental Science and Engineering, Nanjing University of Information Science & Technology, Nanjing 210044, China

[4]Department of Environmental Toxicology, University of California, 1 Shields Ave., Davis, CA 95616, USA

[5]Institute of Low Temperature Science, Hokkaido University, Sapporo 060-0819, Japan

[6]CMA Key Laborotary of Aerosol-Cloud-Precipitation, Nanjing University of Information Science and Technology, Nanjing 210044, China

[7]Aerodyne Research, Inc., Billerica, MA 01821, USA

*Correspondence to*: Yele Sun (sunyele@mail.iap.ac.cn)





**Abstract.** Winter has the worst air pollution in a year in the megacity of Beijing. Despite extensive winter studies in recent years, our knowledge on the sources, formation mechanisms and evolution of aerosol particles is not complete. Here we have a comprehensive characterization of the sources, variations and processes of submicron aerosols that were measured by an Aerodyne high-resolution aerosol mass spectrometer from December 17, 2013 to January 17, 2014 along with offline filter analysis by gas chromatography/mass spectrometry. Our results suggest that submicron aerosols composition was overall similar in winter among different years, which was mainly composed of organics (60%), sulfate (15%) and nitrate (11%). Positive matrix factorization of high- and unit-mass resolution spectra identified four primary organic aerosol (POA) factors from traffic, cooking, biomass burning (BBOA) and coal combustion (CCOA) emissions, and two secondary OA (SOA) factors. POA dominated OA, on average accounting for 56% with CCOA being the largest contributor (20%). Both CCOA and BBOA showed distinct polycyclic aromatic hydrocarbons (PAHs) spectral signatures indicating that PAHs in winter were mainly from coal combustion (66%) and biomass burning emissions (18%). BBOA was highly correlated with levoglucosan, a tracer compound for biomass burning ($r^2 = 0.93$), and made a considerable contribution to OA in winter (9%). An aqueous-phase processed SOA (aq-OOA) that was strongly correlated with particle liquid water content, sulfate and S-containing ions (e.g., $CH_2SO_2^+$) was identified. aq-OOA on average contributed 12% to the total OA and played a dominant role in increasing oxidation degrees of OA at high RH levels (> 50%). Our results illustrate that aqueous-phase processing can enhance SOA production and oxidation states of OA as well in winter. Further episode analyses highlighted the significant impacts of meteorological parameters on aerosol composition, size distributions, oxidation states of OA, and evolutionary processes of secondary aerosols.



## 1 Introduction

Air pollution caused by high concentration of fine particles (PM$_{2.5}$, particles with aerodynamic diameter less than 2.5 µg m$^{-3}$) is of a great concern in densely populated megacities due to its harmful effects to public health (Cao et al., 2012; Madaniyazi et al., 2015). Although the annual average
concentration of PM$_{2.5}$ in Beijing was decreased from 89.5 µg m$^{-3}$ in 2013 to 80.6 µg m$^{-3}$ in 2015, it is still far above the Chinese National Ambient Air Quality Standard (CNAAQS, 35 µg m$^{-3}$ as an annual average). Mitigating air pollution in Beijing remains a great challenge (Zhang et al., 2012; Zhang et al., 2016) not only because of its complex sources and processes, for instance local emissions versus regional transport, and primary emissions versus secondary formation, but also due to the complex
interactions between meteorology and atmospheric aerosols (Ding et al., 2016; Petäjä et al., 2016; Ding et al., 2013). Therefore, understanding of the sources and processes of atmospheric aerosols in the megacity of Beijing is of importance to provide mitigation strategies for the northern pan-Eurasian and Chinese societies related to air quality, and also contributes to the Earth system science and climate policy (Kulmala et al., 2015).

Extensive studies on the basis of various offline and online techniques have been conducted in Beijing in recent years to investigate the concentrations and sources of fine particles. The results show consistently high PM levels in winter than other seasons, mainly due to coal combustion emissions in heating season (Sun et al., 2014; Sun et al., 2013b; Zhang et al., 2013; Liang et al., 2015; Liu et al., 2015). Receptor model analyses, e.g., positive matrix factorization (PMF) and chemical mass balance
(CMB) further confirmed the importance of coal combustion emissions in particulate matter (PM) pollution in winter (Zhang et al., 2013; Zheng et al., 2005). For example, coal combustion aerosol was found to be one of the most important primary components, on average contributing 20 – 33% of the total organic aerosol (OA) mass in winter (Hu et al., 2016; Wang et al., 2015; Sun et al., 2014; Sun et al., 2013b). However, the contributions of traffic emissions are highly uncertain. For instance, Zhang
et al. (2013) reported a contribution of 4% for traffic emissions, whereas Tian et al. (2016) showed a much higher contribution (19%) in Beijing. The reasons for such differences are not clear yet. More recently, many studies highlight the importance of secondary aerosols in the formation of severe haze episodes in megacities (Sun et al., 2014; Yang et al., 2015; Huang et al., 2014; Zhao et al., 2013b; Liu



et al., 2013). Because secondary aerosols are mainly formed over regional scales, its dominance in PM demonstrates the importance of regional transport in the formation of severe haze episodes in north China (Li et al., 2015; Sun et al., 2014; Wang et al., 2014b; Zheng et al., 2015). Such conclusions are further supported by "APEC" studies when secondary aerosol species from regional transport showed

the largest reductions due to emission controls in Beijing and surrounding regions (Sun et al., 2016).

The rapid changes in sources and evolution of severe haze episodes have been also studied using various state-of-the-art online instruments, particularly Aerodyne aerosol mass spectrometer (AMS) that is capable of quantifying size-resolved non-refractory submicron aerosol composition at a high time resolution (Canagaratna et al., 2015). PMF analysis of aerosol mass spectrometer (PMF-AMS)

data illustrated the rapid variations of OA factors from different sources and processes. Primary OA (POA) such as traffic-related hydrocarbon-like OA (HOA), cooking OA (COA), coal combustion OA (CCOA), and biomass burning OA (BBOA) generally showed strong diurnal variations influenced by local source emissions (Huang et al., 2010; Sun et al., 2010a; Sun et al., 2012; Sun et al., 2015; Sun et al., 2013b; Xu et al., 2015; Zhang et al., 2014; Hu et al., 2016; Elser et al., 2016), whereas secondary

aerosols (SOA and SIA) were found to accumulate rapidly in days and contribute most to the severe haze pollution (Sun et al., 2014; Zheng et al., 2015). Recent lab and field studies also indicated that heterogeneous reactions under high RH and $NO_x$ levels might have played important roles in secondary aerosol formation and can increase the PM levels substantially (He et al., 2014; Xie et al., 2015; Quan et al., 2015; Wang et al., 2014a).

Although submicron aerosol composition in Beijing has been relatively well characterized using aerosol mass spectrometry during the last decade (Huang et al., 2010; Sun et al., 2010a; Sun et al., 2012; Sun et al., 2015; Sun et al., 2013b; Xu et al., 2015; Zhang et al., 2014; Hu et al., 2016; Elser et al., 2016), the contributions of primary sources to PM pollution and the evolutionary processes of SOA and its oxidation properties, particularly under high RH conditions remain less understood. For

example, BBOA was mixed within other primary OA factors in previous winter PMF-ACSM analysis due to the limited sensitivity of the ACSM (Sun et al., 2013b; Jiang et al., 2015). The recent two HR-AMS studies also showed substantially different OA composition in different winter in Beijing (Hu et al., 2016; Elser et al., 2016). Elser et al. (2016) found that CCOA contributed approximately 50% of





the total OA mass in $PM_{2.5}$, whereas Hu et al. (2016) reported an average contribution of 24% to OA in $PM_1$. The reasons for such differences are not clear yet although the different size cutoffs might be one of them. In addition, the aqueous-phase processing of OA at low temperatures in winter remains uncertain. While it has been suggested as an important formation pathway of SOA (Ge et al., 2012b;

Lin et al., 2014), Sun et al. (2013a) found that aqueous-phase processing appeared not to significantly enhance SOA and oxidation degree of OA in winter in Beijing. Most importantly, source emissions in northern China are undergoing significant changes since the implementation of "Atmospheric Pollution Prevention and Control Action Plan" on September 10, 2013. As a response, the sources and processes of atmospheric aerosols might have significant changes and vary substantially in different

years. Thus, continuous characterization of aerosol particles in different years in Beijing is of importance for future validating the response of aerosol chemistry to emission controls.

In this study, an Aerodyne high-resolution time-of-flight AMS (AMS hereafter) along with various collocated online instruments were deployed at an urban site in Beijing for real-time characterization of the sources and processes of aerosol particles during winter 2013 - 2014. 3-hour filter samples were

also collected and analyzed for molecular markers with gas chromatography/mass spectrometry (GC/MS). The aerosol composition, size distributions and diurnal variations are comprehensively investigated and also compared with the results in previous studies. The sources of OA are investigated by PMF-AMS analysis of both high mass resolution (HMR) and unit mass resolution (UMR) spectra, together with molecular marker analysis. The evolution processes of OA, particularly

aqueous-phase processing of SOA, and its oxidation degrees are elucidated, and the impacts of meteorological parameters on severe haze formation are demonstrated.

## 2   Experimental methods

### 2.1 Sampling site and instrumentation

The AMS was deployed at an urban site, i.e., Institute of Atmospheric Physics, Chinese Academy

of Sciences, in Beijing that was described in detail in Sun et al. (2012) from December 17, 2013 to January 17, 2014. The sampling site is subject to multiple local influences including cooking emissions from nearby restaurants, traffic emissions from traffic roads and Jingzang highway, and sporadic coal combustion emissions from residential heating. During this study period, ambient





aerosol particles were first drawn inside the sampling room at a flow of 10 L min$^{-1}$, of which ~0.1 was isokinetically sampled into the AMS. The AMS was then operated by alternating the mass-sensitive V-mode and the high mass resolution W-mode every 2 min. The AMS was calibrated for ionization efficiency with 350 nm ammonium nitrate particles following the standard protocols (Jayne et al., 2000; Jimenez et al., 2003), and the relative ionization efficiency (RIE) of ammonium (=5.0) was also determined.

The gaseous species were measured by a range of gas analyzers (Thermo Scientific), including CO (model 48$i$), NO/NO$_y$ (model 42$i$), O$_3$ (model 49$i$), and SO$_2$ (model 43$i$), and the meteorological parameters (winds, temperature and relative humidity) were obtained from the Beijing 325 m meteorological tower nearby.

3-hour PM$_{1.0}$ samples were collected using an air sampler (Zambelli, Italy) at a flow rate of 38.3 L min$^{-1}$ with quartz filters (47 mm diameter, Pallflex), which were pre-combusted at 450°C for 6 h, during January 6 – 9, 2014. Field blanks were collected by placing filters onto the filter holder for a few minutes without pumping before and after the campaign. After sampling, each aerosol or blank filter was wrapped individually with aluminum foil and stored at −18°C in darkness prior to analysis.

**2.2 Chemical analysis**

Aliquots of filter quartz samples were first extracted with dichloromethane /methanol (2:1; v/v), and then concentrated by a rotary evaporator under vacuum after filtration. After dryness with pure nitrogen gas, the extracts were reacted with N,O-bis-(trimethylsilyl)trifluoroacetamide to form derivatives for subsequent GC/MS analysis. In this study, 24 molecular compounds were quantified including sugar compounds (levoglucosan, galactosan, mannosan), 4-hydroxybenzoic acid, vanillin, 2-methyltetrols, and etc. A more detailed filter pretreatment and GS/MS analysis is given in Fu et al. (2008).

**2.3 AMS data analysis**

The AMS data were analyzed for the mass concentrations and size distributions of NR-PM$_1$ species using high resolution data analysis software package PIKA (Sueper, 2016). A collection efficiency (CE) of 0.5 was applied to the entire dataset to compensate for the incomplete detection of





the AMS mainly due to particle bouncing effects (Matthew et al., 2008; Middlebrook et al., 2012). Elemental analysis (EA) was performed on high resolution mass spectra of OA using the software APES (Aiken et al., 2008). The elemental ratios including oxygen-to-carbon (O/C), hydrogen-to-carbon (H/C), nitrogen-to-carbon(N/C), and organic-mass to organic carbon (OM/OC) ratios were

determined with the "Aiken – Ambient" (A-A)(Aiken et al., 2008) method and the recently updated parameterization named "Improved – Ambient" (I-A) (Canagaratna et al., 2015). As shown in Fig. S1, the O/C and H/C ratios are on average 28% and 11% higher than those derived from the A-A method. The elemental ratios reported in this study are obtained from the I-A method unless otherwise stated.

     Positive matrix factorization (Paatero and Tapper, 1994) was performed on high resolution mass

spectra of V-mode and W-mode to retrieve potential OA factors from different sources. Because of the limited mass resolution of AMS, PMF analysis was limited to $m/z$ 150. It should be noted that, for $m/z$ 120 – 150, only major fragment ions were included in the PMF analysis. The data and error matrices were treated according to the procedures detailed in DeCarlo et al. (2010). The PMF solution was then evaluated using a PMF Evaluation Toolkit written in Igor Pro (Ulbrich et al., 2009) following the

procedures described in Zhang et al. (2011). After a careful evaluation of the PMF results, we found that the six factor solution at fpeak = 0 can be well interpreted. The six factors are four POA factors including HOA, COA, BBOA, and CCOA, and two SOA factors, including an oxygenated OA (OOA) and an aqueous-OOA (aq-OOA). As indicated by the diagnostic plot in Fig. S2, the contributions of OA factors were fairly stable across different fPeak values. Figure S3 shows the correlations of six OA

factors with other tracers. It is clear that each OA factor was correlated with specific tracers, for instance, HOA vs. $NO_x$, CCOA vs. PAHs, COA vs. $C_6H_{10}O^+$, BBOA vs. $C_2H_4O_2^+$, OOA vs. $CO_2^+$, and aq-OOA vs. SIA. We also compared the PMF results between V-mode and W-mode. As shown in Fig. S4, the time series of six OA factors were highly correlated between the two modes. In this study, we further performed PMF analysis on UMR mass spectra of V-mode between $m/z$ 12 – 350. Such an

analysis can keep the most important PAHs information (Dzepina et al., 2007) for a better source apportionment despite several recent PMF-HMR analyses to $m/z$ 200 (Hu et al., 2016; Hu et al., 2013). The PMF-UMR solution was also evaluated in the same way as that of PMF-HMR. As indicated in Fig. S5, the mass spectra of six OA factors from PMF-UMR were similar to those from PMF-HMR ($r^2$ = 0.86 - 0.98).





## 2.4 Liquid water content

Liquid water content associated with inorganic species was predicted using ISORROPIA-II model (Nenes et al., 1998; Fountoukis and Nenes, 2007) with AMS aerosol composition and meteorological parameters (temperature and relative humidity) as input. The ISORROPIA-II model then calculated

the composition and phase state of a $K^+$–$Ca^{2+}$–$Mg^{2+}$–$NH_4^+$ –$Na^+$–$SO_4^{2-}$ –$NO_3^-$–$Cl^-$–$H_2O$ in thermodynamic equilibrium with gas-phase precursors.

## 3    Results and discussion

### 3.1 Mass concentrations and compositions

Figure 1 shows the time series of meteorological parameters, gaseous species, and NR-$PM_1$ aerosol

species for the entire study period. The NR-$PM_1$ species varied dramatically throughout the study. The average NR-$PM_1$ mass concentration was 64 ±59 µg m$^{-3}$, which is similar to that observed during winter 2011-2012 (Sun et al., 2013b), yet 32% lower than that measured during the severe pollution month of January 2013 (Zhang et al., 2014). As shown in Fig. 2b, the average NR-$PM_1$ concentrations in Beijing varied from 50 to 94 µg m$^{-3}$ since 2006, and even remained at relatively high levels in 2014

(51 – 67 µg m$^{-3}$ ) after the release of "Atmospheric Pollution Prevention and Control Action Plan" on 10 September 2013. The concentration levels of PM are much higher than the annual CNAAQS  of 35 µg m$^{-3}$, indicating that the air pollution in Beijing is still severe during all seasons (Sun et al., 2015).

Organics constituted a major fraction of NR-$PM_1$, on average accounting for 60% during this study. The dominance of organics is overall consistent with previous winter studies in Beijing, which showed

that 50 – 60% of NR-$PM_1$ was organics (Sun et al., 2013b; Zhang et al., 2015a; Zhang et al., 2014; Sun et al., 2015; Hu et al., 2016). The contribution of organics to NR-$PM_1$ in winter was much higher than that during other seasons (~30 – 40%), which was primarily due to substantial emissions from coal combustion (Sun et al., 2013b; Hu et al., 2016; Elser et al., 2016). This is further supported by a large enhancement of chloride contribution from ~1% in summer to 4 – 6% in winter. Sulfate was the

second largest component (15%) which is comparable to that of nitrate (11%). SIA showed much lower contributions to NR-$PM_1$ in winter compared to summer due to weaker photochemical processing associated with low $O_3$ (5.6 ±7.3 ppb). Aerosol composition of NR-$PM_1$ varied substantially across the entire study, yet it appeared to be strongly affected by RH. While the entire





study period was characterized by low RH (< 40%, 77% of the time), five episodes with RH > 40% were also observed (E1 – E5 in Fig. 1). The periods with high RH levels were clearly characterized by high contributions of SIA, particularly sulfate. Indeed, the sulfate concentration often rapidly exceeded nitrate as the RH increased to above 40%, indicative of aqueous-phase production during wintertime

(Sun et al., 2013a; Sun et al., 2014; Quan et al., 2015). As indicated in Fig. 3a, $SO_4/NO_3$ ratios showed evident increases as a function RH at high PM loading periods (> 30 μg m$^{-3}$). As RH was >40%, the ratios of $SO_4/NO_3$ exceeded one during most of the time, demonstrating a more important role of sulfate in SIA during high RH periods. As a comparison, nitrate was more significant at low RH levels (< 30%). These results elucidate different formation mechanisms in the formation of sulfate and nitrate,

which are mainly driven by aqueous-phase and photochemical production, respectively in winter. We also noticed higher $SO_4/NO_3$ ratios during periods with lower aerosol loadings (< 30 μg m$^{-3}$). One reason is that air masses were dominantly from the north - northwest with much less $NO_x$ emissions compared to the megacities in eastern China (Zhao et al., 2013a). In addition, the shorter life time of nitrate compared to sulfate might also have played a role during the long-range transport from the

north-northwest to Beijing. Figure 3b shows the size-dependence of $SO_4/NO_3$ ratios during five episodes with high RH levels (> 40%) and five episodes with low RH levels (< 40%). It is clear that $SO_4/NO_3$ ratios at high RH levels were consistently higher than those at low RH levels across different size ranges, further supporting the importance of aqueous-phase processing in formation of sulfate. Also, $SO_4/NO_3$ ratios showed clear increases as a function of particle sizes, likely indicating the

different formation processes of sulfate and nitrate in different size ranges. Considering that large accumulation mode particles are generally more aged than smaller particles (Zhang et al., 2005), results here might indicate that nitrate mainly formed via photochemical production played a more important role in SIA at smaller sizes, whereas, sulfate formed over regional scales was more significant in accumulation mode particles.

**3.2 Diurnal cycles and size distributions**

The diurnal cycles varied differently among different NR-PM$_1$ species. As shown in Fig. 4a, organics was characterized by two peaks occurring at noon and evening time. Such a diurnal pattern influenced by cooking emissions at meal times has been observed during all seasons in Beijing (Sun et



al., 2010a; Sun et al., 2012; Sun et al., 2015; Huang et al., 2010; Zhang et al., 2014). High concentration of organics at nighttime was also caused by intensive emissions from other primary sources, e.g., traffic and coal combustion (see Section 3.4 for more detail). The diurnal cycles of sulfate and nitrate were relatively similar, yet largely different from those observed during winter

2011-2012 (Sun et al., 2013b). These two species showed clear increases from ~8:00, and then decreased until late afternoon due to the elevated planetary boundary layer (PBL). Indeed, $SO_4/\Delta CO$ and $NO_3/\Delta CO$ after considering the dilution effect of PBL showed clear increases from 10:00-16:00 (Fig. 4b), indicating daytime photochemical production. Note that the daytime increase of $NO_3/\Delta CO$ (~4 $\mu g\ m^{-3}\ ppm^{-1}$) was nearly twice that of $SO_4/\Delta CO$, suggesting that photochemical processing was

more important for nitrate than sulfate during this study. The primary species (e.g., chloride and PAHs) showed similar diurnal patterns which were both characterized by higher concentrations at nighttime than daytime. Even considering the dilution effect of PBL, PAHs still showed higher concentrations at night demonstrating the increased coal combustion emissions from residential heating.

The size distributions of NR-PM$_1$ species in this study were largely different from those observed

in summer (Hu et al., 2016; Huang et al., 2010; Sun et al., 2010a), yet resembled to those in winter (Hu et al., 2016). While all NR-PM$_1$ species showed similarly large accumulation modes peaking at ~600 nm in summer (Huang et al., 2010; Sun et al., 2010a), they varied differently in winter. Organics showed a broad size distribution peaking at ~450 nm. Organics dominated NR-PM$_1$ at small sizes with a contribution as high as 90%, indicating that a substantial fraction of ultrafine particles during

wintertime was organics. Indeed, organic particles below 100 nm showed the best correlation with COA (Fig. S6), indicating a dominant contribution of cooking emissions on ultrafine particles. The contribution of organics decreased rapidly as a function of size, and larger organic particles (e.g., > 400 nm) correlated much better with SOA and SIA (Fig. S6). These results indicate the different sources in organics at different size ranges. While small organic particles were mainly from primary

emissions, those at large size ranges were primarily contributed by secondary aerosols. The size distributions of nitrate and sulfate were quite different. While the two species were likely internally mixed with similar size distributions in summer, they peaked at different sizes which are ~600 nm and ~300 – 400 nm, respectively. These results indicate that sulfate and nitrate were more likely externally mixed and have different formation mechanisms in winter. Indeed, although nitrate was correlated



well with sulfate ($r^2 = 0.81$), the correlations and the ratios of $SO_4/NO_3$ were strongly RH dependent, indicating the different roles of aqueous-phase and photochemical processing in the formation of sulfate and nitrate. While nitrate showed comparable contributions across different size ranges (~10 – 15%), the sulfate contribution was significantly elevated at larger sizes with a contribution up to 30%.

PAHs showed a large single accumulation mode peaking at ~400 nm. The contribution of PAHs also peaked at similar size ranges indicating that coal combustion emissions appeared not to emit a large amount of small particles.

**3.3 Oxidation states of OA**

  Figure 5 shows the time series of H/C and O/C ratios during this study period. The O/C varied

substantially with hourly average ranging from 0.18 to 0.71. The average O/C was 0.37±0.10 (0.29 ±0.08 with A-A method), which was relatively low compared to those (0.32 – 0.41, A-A method) previously reported in Beijing (Huang et al., 2010; Zhang et al., 2015a; Zhang et al., 2014; Xu et al., 2015). These results suggest that OA in this study was overall less oxidized. However, several periods with high O/C ratios, e.g., January 6 and January 17, associated with high RH were also observed,

indicating very different oxidation degrees of OA. The OM/OC ratio was highly correlated with O/C ($r^2 = 0.998$) with an average value of 1.64±0.13 (1.53±0.11 with A-A method). The OM/OC ratio is similar to that suggested for the urban sites (Turpin and Lim, 2001). The O/C ratio showed a pronounced diurnal cycle with higher values during daytime. Similar O/C diurnal cycles have been widely observed in various seasons in Beijing (Zhang et al., 2014; Zhang et al., 2015a; Hu et al., 2016).

However, the daytime increase of O/C was interrupted by a clear decrease at noon time due to the influences of cooking emissions. In fact, the average O/C ratio by excluding the COA contribution showed a continuous increase from 0.38 at 8:00 to 0.55 at 16:00, and the H/C ratio showed a corresponding decrease from 1.72 to 1.64. Such a diurnal cycle demonstrated the photochemical processing in daytime. The Van Krevelen plot showed that the relationship between H/C and O/C

appeared to be quite different at different levels of oxidation. For example, the slope of H/C vs. O/C was -0.70 during the period with the COA contribution larger than 40%, which is much steeper than -0.42 during the period with high aq-OOA (> 20%). These results indicated the different evolutionary mechanisms between primary OA and highly oxidized OA (Chen et al., 2015).





Figure 6a shows the variations of O/C as a function of RH. The O/C first showed a slight decrease at RH < 50%, and then had a rapid increase until RH = 80%. The O/C ratio by excluding the influences of COA showed similar RH-dependence. Higher O/C ratios at higher RH levels likely indicate the aqueous-phase processing in the formation of highly oxidized organic aerosols. By investigating the OA composition change as a function of RH, we found that the POA contribution decreased from ~60% to 25% as the RH increased from 40% to 80%, correspondingly, the SOA contribution increased from 40% to 70%, and in particular aq-OOA from 10% to 40%. As a result, the increase of O/C ratio at high RH levels was mainly caused by the increase of aq-OOA. As discussed in section 3.4.5, aq-OOA in this study is likely an oxidized aqueous-phase processed OA. These results suggest that aqueous-phase processing can form highly oxidized SOA and enhance oxidation degree of OA as well in winter (Ge et al., 2012b). This conclusion appeared to be different from that in a previous winter study in which $f_{44}$ (fraction of $m/z$ 44 in OA), a surrogate of O/C, was relatively constant at high RH levels (Sun et al., 2013a). We noticed that the variations of mass concentrations of OA factors as a function of RH between the two studies. For example, most OA factors showed almost linear increases at low RH levels reaching maxima at ~50%, and then decreased rapidly between 50%-80% (Fig. S7). The aq-OOA concentration was low at RH < 40%, and showed a large increase between 40-60%, then remained relatively constant at high RH levels. Such a RH-dependence of OA composition was significantly different from that observed during winter 2011-2012 when all OA factors remained high concentrations at high RH levels (Sun et al., 2013a). Our results suggest that the OA sources and evolution processes at high RH levels might vary significantly in different years.

## 3.4 Sources and variations of OA

Compared to our previous ACSM study during winter 2011-2012 (Sun et al., 2013b), PMF-HMR and PMF-UMR were able to identify six OA factors, including four POA factors (HOA, COA, BBOA, and CCOA) and two SOA factors (OOA and aq-OOA). Although similar PMF resolution was obtained in winter 2010 and 2014 (Hu et al., 2016; Elser et al., 2016), we found that OA sources, variations and processes can vary substantially in different years.

### 3.4.1 Hydrocarbon-like OA



The HOA spectrum was characterized by typical hydrocarbon ion series of $C_nH_{2n+1}^+$ and $C_nH_{2n+1}^+$ (Fig. 7a), which is similar to those observed at various urban sites (Ng et al., 2011) and diesel exhausts (Canagaratna et al., 2004). Consistently, HOA was highly correlated with $C_nH_{2n+1}^+$ ions, e.g., $C_3H_7^+$, $C_4H_9^+$, $C_5H_{11}^+$, $C_6H_{13}^+$, $C_7H_{15}^+$, and $C_8H_{17}^+$ ($r^2 > 0.80$, Fig. 8). The O/C of HOA is 0.11, which is lower

than that observed in summer (0.17) (Huang et al., 2010) and January 2013 (Zhang et al., 2014), indicating that HOA during this study period was primarily from fresh emissions. Indeed, HOA was tightly correlated with $NO_x$ and CO ($r^2 = 0.73$ and 0.69, respectively, Fig. S3), two tracers for vehicle emissions, yet presented much weaker correlations with secondary aerosol species ($r^2 < 0.3$). The diurnal cycle of HOA was characterized by high concentration at night due to the enhanced traffic

emissions from diesel trucks and heavy duty vehicles (Han et al., 2009), and also shallow PBL at night. Such a diurnal cycle was similar to that previously observed in winter in Beijing (Sun et al., 2013b; Zhang et al., 2015a; Hu et al., 2016). HOA on average accounted for 10% (6 – 14%) of the total OA for the entire study period, which is much lower than 18% during the Beijing 2008 Olympic Games (Huang et al., 2010), yet close to 11% reported in January 2013 (Zhang et al., 2014) and 14% in winter

2010 (Hu et al., 2016).

### 3.4.2    Cooking OA

Similar to previously reported COA, the mass spectrum was characterized by high *m/z* 55/57 ratio (Sun et al., 2011; Mohr et al., 2012; He et al., 2010). COA contributed 31% and 36% to $C_3H_3O^+$ and $C_4H_7^+$, respectively at *m/z* 55, and ~21% to $C_3H_5O^+$ and $C_4H_9^+$ at *m/z* 57. Although *m/z* 55 is often

used a tracer for cooking emissions, our results indicated that COA correlated much better with several other fragment ions, e.g., $C_6H_{10}O^+$, $C_5H_8O^+$, $C_4H_6^+$, $C_5H_7^+$, and $C_6H_8^+$ (Fig. 8). Compared to the results in New York City, COA showed similarly tight correlations with $C_6H_{10}O^+$ and $C_5H_8O^+$, yet much weaker correlations with large hydrocarbon ions (Sun et al., 2011). These results suggest that $C_6H_{10}O^+$ and $C_5H_8O^+$ are better tracers for cooking emissions than the unit *m/z* 55 and 57 although

they contribute small fractions to the total organics. The ratio of COA/$C_6H_{10}O^+$ is 184, which is close to that obtained in winter in Fresno, California (Ge et al., 2012a) and New York City (Sun et al., 2011) (170 and ~180, respectively). As a result, the COA aerosol could be simply estimated with the equation of COA = $180 \times C_6H_{10}O^+$. The O/C of COA is 0.14 which is similar to those observed in





winter in Beijing and Fresno, CA (Ge et al., 2012a; Zhang et al., 2014; Hu et al., 2016), and fresh cooking emissions (0.08 – 0.13) (He et al., 2010). These results indicate that COA is mainly composed of low oxygenated organics. In addition, COA showed much better correlations with smaller organic particles, e.g., < 100 nm (Fig. S6), indicating substantial emissions of ultrafine particles from cooking activities.

COA showed a pronounced diurnal cycle with two prominent peaks at lunch and dinner times, and a visible morning breakfast peak at ~8:00. Such a unique diurnal cycle of COA has been observed many times in megacities. COA on average accounted for 18% of the total OA with the highest contribution close to 40% during meal times. COA has been found to show comparable and even higher contributions than traffic-related HOA in densely-populated megacities, e.g., New York City (16% vs. 14%) (Sun et al., 2011), Paris (17% vs. 11-13%) (Crippa et al., 2013), London (22-30% vs. 23 – 25%) (Allan et al., 2010), Fresno (19% vs. 22%) (Ge et al., 2012a), and Beijing (24% vs. 18%) (Huang et al., 2010). A recent study found that emissions from residential heating and cooking have the largest impact on premature mortality globally (Lelieveld et al., 2015). Particularly, aerosol particles tend to present at small size ranges during the periods with high COA contribution, which are subject to have more adverse health effects. Therefore, reducing cooking emissions is of importance not only in mitigating urban PM pollution, but also alleviating the harmful effects in densely populated megacities.

### 3.4.3 Coal combustion OA

CCOA showed a similar spectral pattern to HOA at small $m/z$'s (< 120) (Fig. 7d). The largest difference is $m/z$ 115 (mainly $C_9H_7^+$) which is prominent in CCOA spectrum yet much smaller in HOA spectrum. Although CCOA was separated from HOA by PMF analysis of ACSM UMR spectra in our previous study during winter 2011-2012 (Sun et al., 2013b), the interpretation of these two components were quite difficult due to the absence of the information at large $m/z$'s (> 150). In this study, PMF analysis of UMR spectra to $m/z$ 350 showed strong PAH signatures in CCOA spectrum, which are $m/z$'s 152, 165, 178, 189, 202, 215, 226, 239, 252, 276, 300, 326, and 350 (Dzepina et al., 2007). The distinct PAH peaks of $m/z$'s 152, 165, 178, and 189 were already observed in CCOA spectrum resolved in Beijing in winter 2010 (Hu et al., 2016) and a rural site in central China (Hu et al.,



2013). Comparatively, the traffic related HOA did not present pronounced PAH signals, which was different from those observed at morning rush hours in Mexico City (Dzepina et al., 2007). This likely indicates that a dominant source of PAHs is coal combustion. In fact, CCOA was highly correlated with PAHs ($r^2$ = 0.92, Fig. 8d). In addition, CCOA was found to have ubiquitously tight correlations with large $m/z$'s (> 150) (Fig. S8), and contributed more than 40% for most $m/z$'s (Fig. S9). This likely indicates that coal combustion emits a considerable amount of high molecular weight organics. The O/C of CCOA is 0.14 suggesting that organic aerosols emitted from coal combustion are fresh. Higher O/C ratio of CCOA (0.17) observed at the rural site (Hu et al., 2013) was likely due to the atmospheric aging during the transport.

The average mass concentration of CCOA was 7.6 μg m$^{-3}$, which is lower than that (11.3 μg m$^{-3}$) observed during winter 2011-2012 (Sun et al., 2013b), yet similar to those (7.3 – 8.2 μg m$^{-3}$) in January 2013 and winter 2010 (Sun et al., 2014; Hu et al., 2016). CCOA was the largest primary OA, on overage accounting for 20%. CCOA showed a pronounced diurnal cycle with significantly higher concentration at nighttime than daytime (Fig. 8g).The contribution of CCOA to OA reached 37% at nighttime, which is much higher than 7% during daytime. After considering the PBL dilution effect, CCOA still showed a pronounced diurnal variation, illustrating a much stronger coal combustion emissions at night.

### 3.4.4  Biomass burning OA

Compared to previous AMS winter studies in Beijing (Zhang et al., 2015a; Zhang et al., 2014; Sun et al., 2014; Sun et al., 2013b), we were able to resolve a factor, the mass spectrum of which was characterized by prominent $m/z$ 60 (mainly $C_2H_4O_2^+$) and 73 ($C_3H_5O_2^+$), two markers indicative of biomass burning emissions (Mohr et al., 2009; Lanz et al., 2007). The two recent AMS studies also resolved BBOA factors in winter Beijing (Hu et al., 2016; Elser et al., 2016). Consistently, the BBOA spectrum resembles to those observed at various sites (Ng et al., 2011), e.g., Fresno, CA (Ge et al., 2012a) and Mexico City (Aiken et al., 2009). Although BBOA was tightly correlated with $C_2H_4O_2^+$ ($r^2$ = 0.65), similar high correlations between $C_2H_4O_2^+$ and primary HOA and CCOA were also observed indicating the contributions of multiple combustion sources to $m/z$ 60 (Fig. 8). Indeed, the four primary OA factors showed comparable contributions to ion $C_2H_4O_2^+$ in this study. We noticed that





BBOA showed the best correlation with $C_6H_6O_2^+$ (*m/z* 110), a marker ion for hydroquinone and catechol from the pyrolysis of lignin and/or the hydroxylation of phenol (Sun et al., 2010b), supporting the influence of biomass burning emissions on this factor. In addition, similar strong PAHs signals to CCOA were also observed at large *m/z*'s suggesting that BBOA is also an important source of PAHs

during wintertime. BBOA showed a similar diurnal pattern to CCOA which was characterized by high concentration at nighttime. The O/C ratio of BBOA is 0.36, which is similar to those observed at other urban sites (Aiken et al., 2009; Ge et al., 2012a; Huang et al., 2011; He et al., 2011). It's interesting to note that BBOA presents a high N/C ratio (=0.042) compared to the other three primary factors which is likely due to the large amount of nitrogen-containing organic compounds emitted from biomass

burning emissions (Laskin et al., 2009). Consistently, high N/C ratios in BBOA (~0.06) were also observed in Pearl River Delta (He et al., 2011; Huang et al., 2011). The average mass concentration of BBOA was 3.3 μg m$^{-3}$ (9% of the total OA), indicating that BBOA is an important source of OA during wintertime. The concentration and contribution of BBOA was overall consistent with the values previously reported in Beijing in winter (Hu et al., 2016; Elser et al., 2016).

BBOA was highly correlated with levoglucosan (r$^2$ = 0.93, Fig. 10a), a common tracer compound for biomass burning (Simoneit, 2002). The average ratio of levoglucosan to organic carbon (OC) derived from BBOA is 0.257, which is generally higher than those from burning of individual biomass (Sullivan et al., 2008), and that (0.036) during the BB episode in summer in Beijing (Cheng et al., 2013). Previous studies showed that coal combustion also emits a considerable amount of

levoglucosan (Zhang et al., 2008). In fact, levoglucosan was also correlated with the total OC from BBOA and CCOA (r$^2$ = 0.91), yielding an average ratio of levoglucosan /OC of 0.062. The ratio of levoglucosan/mannosan (*L/M*) can be used to indicate different types of burning. For example, burning of hard wood showed much higher *L/M* ratio compared to that of softwood. Cheng et al. (2013) observed quite different *L/M* ratios for different types of biomass burning, e.g., 12.7 for wheat straw

and 19.7 for corn straw, yet much lower values for pine and poplar wood. In this study, levoglucosan was highly correlated with mannosan (Fig. 10b, r$^2$ = 0.95), and the ratio of *L/M* (17.1) is similar to those from burning of wheat straw and corn straw (Cheng et al., 2013). We also noticed that the *L/M* ratio in this study is similar to those (14.8 – 19.6) from residential burning of bituminite and coal briquette (Zhang et al., 2008), likely indicating the contribution of coal combustion to these two




compounds. This is further supported by the visible *m/z* 60 in CCOA spectrum. As a result, levoglucosan and mannosan observed in this study were likely from both biomass burning and coal combustion, however it is difficult to separate them only based on *L/M* ratios.

### 3.4.5   Secondary organic aerosols

Two SOA factors, i.e., OOA and aq-OOA, with the similar spectral patterns, yet largely different time variations were identified. While the O/C ratios were similar between OOA and aq-OOA (0.75 vs. 0.81), the H/C ratio was much different (1.51 for OOA and 1.75 for aq-OOA). We also noticed the different ratios of $C_2H_3O^+/CO_2^+$ and $CHO^+/CO_2^+$ in OOA and aq-OOA spectra. Indeed, the correlations between OOA/aq-OOA and individual fragment ions were quite different. As shown in
Fig. 8, OOA was highly correlated the oxygenated ions series $C_xH_yO_2^+$ and $C_xH_yO_1^+$, e.g., $CO_2^+$ (*m/z* 44, $r^2 = 0.89$), $C_2H_3O^+$ (*m/z* 43, $r^2 = 0.85$), $C_3H_2O_2^+$ (*m/z* 70, $r^2 = 0.90$), and $C_4H_2O_2^+$ (*m/z* 82, $r^2 = 0.85$) etc., yet the correlations with hydrocarbon ions $C_xH_y^+$ were much weaker. These results indicate that OOA was primarily composed of oxygenated organics. OOA was tightly correlated with $O_x$ ($r^2 = 0.73$) and $NO_3$ ($r^2 = 0.71$), consistent with previous results observed in January 2013 (Sun et al., 2014). The
diurnal cycle of OOA was significant showing an increase from 8:00 until 20:00. The daytime increase was interrupted by a temporary decrease in the afternoon due to the dilution effect of PBL. In fact, OOA/ΔCO presented a continuous increase from 8:00 to 14:00, and then remained at a relatively high level until 19:00 (Fig. S10). These results clearly indicate the photochemical production of OOA in daytime. Based on the variations of OOA between 8:00 – 14:00, we estimated that local
photochemical production could contribute ~70% of OOA during this study. OOA on average contributed 25% of the total OA with the contribution as high as 42% in the late afternoon.

The aq-OOA was highly correlated with specific unique fragment ions from typical aqueous-phase processing products. For example, aq-OOA correlated well with $C_2H_2O_2^+$ (*m/z* 58), $C_2O_2^+$ (*m/z* 56), and $CH_2O_2^+$ (*m/z* 46) (Fig. 8). These ions are typical fragment ions of glyoxal and methylgloxal
(Chhabra et al., 2010), which are important precursors in the formation of low volatility SOA in cloud processing (Carlton et al., 2007; Altieri et al., 2008; Tan et al., 2009). In addition, aq-OOA was also highly correlated with several sulfur-containing ions, e.g., $CH_3SO^+$, $CH_2SO_2^+$, and $CH_3SO_2^+$ (Fig. 8), which are typical fragment ions of methanesulfonic acid (MSA), a secondary product from the



oxidation of DMS (Ge et al., 2012b; Zorn et al., 2008). Previous studies have found that hydromethanesulfonate (HMS) can be used a tracer for aqueous-phase fog processing. Although the HMS spectrum does not present $CH_2SO_2^+$, and $CH_3SO_2^+$ peaks, it is unique in the absence of $SO_3^+/HSO_3^+$ and $H_2SO_4^+$ compared to pure ammonium sulfate (Ge et al., 2012b). Therefore, an

elevated $SO_2^+/SO_3^+$ ratio would be expected if there is a considerable aqueous-phase formation of HMS. In this study, we did observe a clear increase of the $SO_2^+/SO_3^+$ ratio as a function of RH (Fig. S11) while the $SO^+/SO_2^+$ ratio remained relatively constant (~0.69) across different RH levels. This result indicates the importance of aqueous-phase processing in this study. In addition, a high N/C ratio (0.045) was also observed for aq-OOA, which is consistent with recent findings that aqueous-phase

processing of glyoxal/methylglyoxal with amino acids and amines can form nitrogen-containing compounds (De Haan et al., 2009; De Haan et al., 2010; Ge et al., 2011) .

      The time series of aq-OOA was largely different from other OA factors. The variations of aq-OOA tracked well with RH. As shown in Fig. 9f, aq-OOA showed high concentrations at high RH levels (> 40%), while remained at low levels at RH < 40%. The contribution of aq-OOA to OA showed a strong

RH-dependence. While it contributed less than 10% to OA at RH < 40%, the contribution rapidly increased as a function of RH, and reached 44% at RH = 80% (Fig. S7b). We further compared the time series of aq-OOA with aerosol liquid water content (LWC, Fig. 9f). It is clear that the periods with high LWC were characterized by high aq-OOA. These results support that aq-OOA is a secondary factor that was strongly associated with aqueous-phase processing. Further support is the

tight correlation between aq-OOA and sulfate ($r^2$ = 0.93), a species primarily from aqueous-phase/cloud processing. We also noticed relatively high concentration of aq-OOA during some periods with low RH and LWC. Considering that aq-OOA showed relatively flat diurnal variations, these results might indicate that aq-OOA is also likely from regional transport during which it was well processed via either photochemical processing or aqueous-phase processing.

The SOA (= OOA + aq-OOA) on average accounted for 44% of OA, which is higher than that (31%) reported during wintertime 2011-2012 (Sun et al., 2013b), yet lower than that (54-55%) observed in January 2013 with severe PM pollution (Sun et al., 2014; Zhang et al., 2014). OOA and



aq-OOA on average contributed 32% and 12%, respectively to OA, which indicated that photochemical production was the major process in the formation of SOA during this study period.

**3.5 Polycyclic aromatic hydrocarbons (PAHs)**

PAHs mainly from incomplete combustion are of a great concern in megacities due to their

carcinogenicity and mutagenicity (Boström et al., 2002). Knowledge of their concentration levels and sources is thus important for mitigation strategies of air pollution. Here we quantified the PAHs with AMS using the algorithm developed by Dzepina et al. (2007). The average concentration of PAHs is 0.22 (±0.27) μg m$^{-3}$ for the entire study. The concentration is significantly higher than that (21 – 29 ng m$^{-3}$) reported in summer in Lanzhou (Xu et al., 2014) and in autumn in Beijing (Zhang et al., 2015b),

yet close to that observed in the heating season (Okuda et al., 2006). These results illustrate the largely different sources of PAHs between heating season and non-heating season. In this study, PAHs presented a pronounced diurnal cycle with significantly higher concentration at nighttime than daytime. Such a strong diurnal variation is remarkably similar to that of CCOA, confirming that coal combustion is a large source of PAHs during wintertime (Zhang et al., 2008). Indeed, the mass

spectrum of CCOA showed distinct fragment $m/z$'s from PAHs, e.g., $m/z$ 152, 165, 178, 189, 202, 215, 226, 239, 252, 276, 300, 326, etc. (Fig. 7d). Note that the mass spectrum of BBOA (Fig. 7c) was also characterized by similar PAHs $m/z$'s suggesting that BBOA might also be a considerable contribution. For example, PAHs were highly correlated with BBOA in Fresno, California, yet the correlations with COA and HOA were much weaker. Similar tight correlations between BBOA and PAHs were also

observed during the APEC summit in Beijing (Zhang et al., 2015b). For a better understanding of the sources of PAHs, a linear regression analysis of OA factors was performed on PAHs.

[PAHs] = $a_{CCOA}$ × [CCOA] + $a_{BBOA}$ × [BBOA] + $a_{HOA}$ × [HOA] + $a_{COA}$ × [COA]+ $a_{other}$ × [other] (1)

The average contributions of OA sources to PAHs are shown in Fig. 11a. Coal combustion is the

dominant source of PAHs, on average accounting for 66%. This is consistent with the results from previous studies that coal combustion emits a large amount of PAHs (Zhang et al., 2008; Chen et al., 2005). Biomass burning and traffic emissions contributed 18% and 11%, respectively to the total PAHs. Cooking emissions were a minor source of PAHs in this study, on average accounting for 3%,




which is also consistent with the fact that the amount of PAHs emitted from the Chinese cooking was small (Zhao et al., 2006).

Fig. 11b shows a comparison of OA spectra during a haze episode on December 24 with that during rush traffic hours in Mexico City (Dzepina et al., 2007). While the spectra were both

characterized by distinct PAH signals, the PAH spectra patterns were quite different, indicating that the composition of PAHs was substantially different between coal combustion and traffic emissions. We further compared the OA spectra at two different nighttime periods, and the spectra pattern ($m/z >$ 150) was remarkably similar ($r^2 = 0.99$, Fig. 11d), suggesting similar source emissions in different days. However, the average OA spectra in the afternoon showed differences from that at nighttime,

likely indicating that photochemical processing changed OA composition to a certain degree. The average size distribution of PAHs showed a large single mode peaking at ~400 nm, which is similar to that observed in Mexico City (Dzepina et al., 2007). The size distributions of PAHs were quite similar between different episodes which were both characterized by single accumulation modes peaking at ~400 – 500 nm.

**3.6 Episode analysis**

Many previous studies have found largely different aerosol composition between clean periods and haze episodes in Beijing (Huang et al., 2010; Sun et al., 2012; Sun et al., 2013b; Jiang et al., 2015). In this study, we also observed frequent changes of clean periods and pollution episodes (Fig. 1). Five episodes (E1-E5) with relatively high RH levels (> 50%), five episodes (M1 – M5) with RH levels

between 20 – 40%, and two clean periods (C1-C2) were selected to investigate the variations of aerosol chemistry among different episodes.  The average mass concentrations of NR-PM$_1$ varied substantially from 105 to 255 µg m$^{-3}$ during E1 – E5, which was generally higher than those (67 – 158 µg m$^{-3}$) during M1 – M5. The average composition was also different. While the contribution of organics varied between 40 – 58% during E1 – E5, it was relatively constant at 62 – 67% during M1 –

M5, indicating a more important role of organics in PM pollution at lower RH levels. Sulfate also played a very different role during the two types of episodes with a much higher contribution (17 – 24%) during E1 – E5 than 8 – 13% during M1 – M5.  This result is consistent with our previous conclusion that sulfate played an enhanced role at high RH levels due to aqueous-phase processing.





OA composition varied greatly among different episodes. But overall, episodes with high RH levels showed much higher aq-OOA contributions than those in episodes with low RH levels. Because the average temperature and wind speed were relatively similar, such compositional differences were mainly caused by different RH conditions. For instance, two episodes, i.e., E3 and E5 showed much

high SOA contributions (67 – 77%) than the other episodes (22 – 49%). The two episodes of E3 and E5 were characterized by the highest RH and aq-OOA (37 – 45%), likely indicating strong aqueous-phase processing of OA. The O/C ratios during the two episodes were also the highest (0.56 – 0.60), demonstrating that aqueous-phase processing enhanced the oxidation stages of OA. CCOA was the largest primary OA for most episodes with the contributions ranging from 22 – 31% except E3 and E5.

Several episodes (M3 and C1 – C2) with significant local cooking influences (COA: 27 – 45%) were also observed. In addition to the RH impacts, the sources emissions were also likely different among different episodes. The CO/$NO_x$ ratios varied from 23 to 40 during E1 – E5, while they were generally low ranging from 15 – 22 during M1 – M5. Also, the $NO_2$/$NO_x$ ratio were generally high (0.40 – 0.59) during E1 – E5 than 0.33 – 0.45 during M1 – M5, and even higher (0.67 – 0.69) during C1 – C2. The

size distributions of aerosol species are also substantially different between the two types of episodes. The episodes with high RH levels (E1 – E5) showed larger maximum diameters than those during M1 – M5, for instance, > 500 nm vs. 300 – 500 nm for sulfate, and 400 – 700 nm vs. 200 – 400 nm for nitrate. In contrast, the differences in size distributions of organics and PAHs were much smaller (Fig. 12b). Larger particles of sulfate and nitrate were likely due to the hygroscopic growth at high RH

levels.

Figure 13 shows the evolution of secondary aerosol species and size distributions during two periods with different RH levels. The first period contains two episodes, i.e., M1 and M2 in Fig. 1. The formation of M1 was clearly associated with a change of air masses from the north to the southwest that occurred at approximately 12:00 on December 21. Winds remained consistently from the

southwesterly in the next 10 hours and RH increased gradually from ~20% to 40%. Under such meteorological conditions, nitrate and sulfate showed gradual increases from less than 5 µg m$^{-3}$ to more than 20 µg m$^{-3}$. The mean geometric diameters (GMD) showed similar increases, approximately from 180 nm to 450 nm for nitrate, and 270 nm to 570 nm for sulfate. However, we observed different evolution behaviors between OOA and aq-OOA. While OOA showed similar increases to sulfate and





nitrate, aq-OOA remained at relatively low concentrations. This indicates a dominance of OOA over aq-OOA during the early stage of regional transport with low RH levels. The episode of M1 was rapidly cleaned by the northerly winds until 12:00 on December 22 when wind direction was changed to the south-southwest and the episode of M2 was formed. It is interesting to note that the wind direction changes did not result in immediate increases of all secondary aerosol species. In fact, wind speed was firstly decreased at both ground level and higher altitudes (e.g., 320 m) associated with corresponding decreases of RH. Nitrate and OOA showed clear increases during this stage, while the sulfate and aq-OOA concentrations were consistently low. One explanation is that secondary aerosols during the early stage of these two episodes were mainly from local photochemical production, consistent with the increasing O/C ratios. After wind speed increased significantly at higher altitudes, relatively more humidified air masses from the south-southwest arrived at the sampling site, leading to dramatic increases of all secondary aerosol species.

The formation of Episode E4 was more rapid compared to M1 (Fig. 13b). The mass concentrations of secondary sulfate, nitrate, and OOA showed dramatic increases by a factor of 2 − 5 in one hour (17:00 − 18:00) from 11 to 24 μg m$^{-3}$, 6 to 30 μg m$^{-3}$, and 15 to 52 μg m$^{-3}$, respectively. The size distributions of sulfate and nitrate also showed sudden changes with the maximum diameters increasing from 370 to 600 nm and 260 to 500 nm, respectively. The only explanation for such rapid changes is regional transport. As indicated in Fig. 14, the air mass trajectories changed from the northwest to the south by circulating through the west during the formation stage of E4. The MODIS satellite image indicated a clear dividing line of clean air and haze between the west-northwest and south-east, while the sampling site in the city is located on the line. Such a spatial distribution of haze was mainly caused by the topography of the north China plain where Mount Taihang is in the west and Mount Yan is in the north. No doubt, the PM levels and composition can have significant changes when air masses switched between the two sides of the diving line. For example, the evolution of E4 was interrupted by a half-day clean period that was strongly associated with the northwestern air masses (blue lines in Fig. 14). But the air pollution was changed back when air masses were switched to the southeast (orange lines in Fig. 14). While the concentration levels and size distributions of sulfate and nitrate remained relatively constant during E4, SOA showed substantial changes. OOA dominated SOA during the early stage of E4, yet aq-OOA increased gradually and was comparable to





OOA. Such a change of SOA was consistent with the variations of RH and LWC. As RH and LWC increased, the contributions of aq-OOA increased correspondingly and even exceeded OOA during E5. This further demonstrated an increasing role of aq-OOA in PM pollution at high RH levels in winter. Consequently, OA became more oxidized during the evolution with O/C ratios varied from ~0.4 to 0.8.

**4  Conclusions**

We have a comprehensive characterization of submicron aerosol composition, size distributions, sources and processes of OA in the megacity of Beijing during winter 2013 – 2014. Submicron aerosol species varied dramatically across the entire study largely due to meteorological changes, yet the bulk composition was consistent with previous winter studies, which was mainly composed of organics

(60%), sulfate (15%) and nitrate (11%). The size distributions varied differently among different aerosol species, particularly sulfate and nitrate peaked differently at ~600 nm and ~300 – 400 nm. The different diurnal cycles and RH dependence of sulfate and nitrate further illustrated their different formation mechanisms in winter, which is mainly driven by aq-phase processing and photochemical production, respectively.

The oxidation properties, sources and processes were investigated with elemental analysis and PMF. While the relatively low O/C ratio suggested the less oxidized properties of OA, the diurnal cycle by excluding the COA influence showed a clear increase from 0.38 to 0.55 during daytime, indicating photochemical production of SOA. In addition, we observed an evident increase of O/C as a function of particle liquid water content at high RH levels (> 50%). Such a RH dependence of O/C

was mainly caused by the increase of aq-OOA at high RH levels. These results indicate that aqueous-phase processing enhanced SOA production and oxidation states of OA as well in winter. PMF analysis of OA identified four primary sources, i.e., traffic, cooking, biomass burning, and coal combustion, and two secondary factors. CCOA was the largest contributor of POA, on average accounting for 20%, followed by COA (18%). The CCOA spectrum showed distinct PAH signatures

and was high correlated with PAH ($r^2$ = 0.92). BBOA showed a tight correlation with the tracer compound levoglucosan ($r^2$ = 0.93). The average levoglucosan/BBOA ratio is 0.169 for the entire study, which might be used to estimate BBOA concentrations using molecular marker levoglucosan.



The two SOA factors showed largely different time and diurnal variations. While OOA was highly correlated with photochemical–processing products (e.g., $O_x$, $NO_3$ and fragment ions of $CO_2^+$), aq-OOA was strongly correlated with aq-phase processed products (e.g., liquid water content, sulfate, and S-containing ions, e.g., $CH_2SO_2^+$, and $CH_3SO_2^+$). This result illustrated two different formation

mechanisms in formation of SOA in winter. Although POA dominated OA for the entire study (56%), episode analyses highlighted a more important role of SOA in OA (67 – 77%) at high RH levels. Aerosol composition and size distributions can vary largely among different episodes not only caused by different meteorological conditions, but also due to the air masses from different source regions. Particularly, the switching of air masses between northwest and south-southeast can lead to the rapid

formation and cleaning of haze episodes.

**Acknowledgements**

This work was supported by the National Key Basic Research Program of China (2014CB447900; 2013CB955801), the National Natural Science Foundation of China (41575120; 41175108; 41475117), and the Strategic Priority Research Program (B) of the Chinese Academy of Sciences (XDB05020501).

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





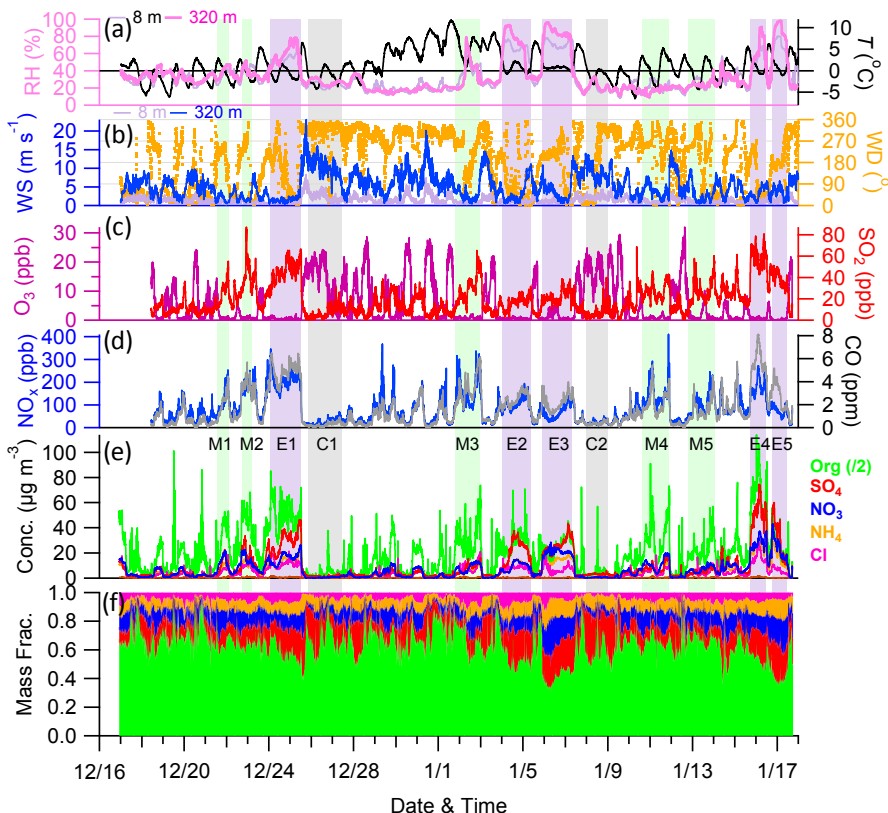

**Figure 1.** Time series of (a) relative humidity (RH) and temperature (T), (b) wind speed (WS) and wind direction (WD), (c) $O_3$ and $SO_2$, (d) CO and $NO_x$, (e) mass concentrations of NR-PM$_1$ species, and (f) mass fractions of NR-PM$_1$ species for the entire study period. In addition, five episodes with

5   relatively high RH levels (E1-E5), five episodes with moderately high RH levels (M1-M5), and two clean periods (C1, C2) are marked for further discussions.


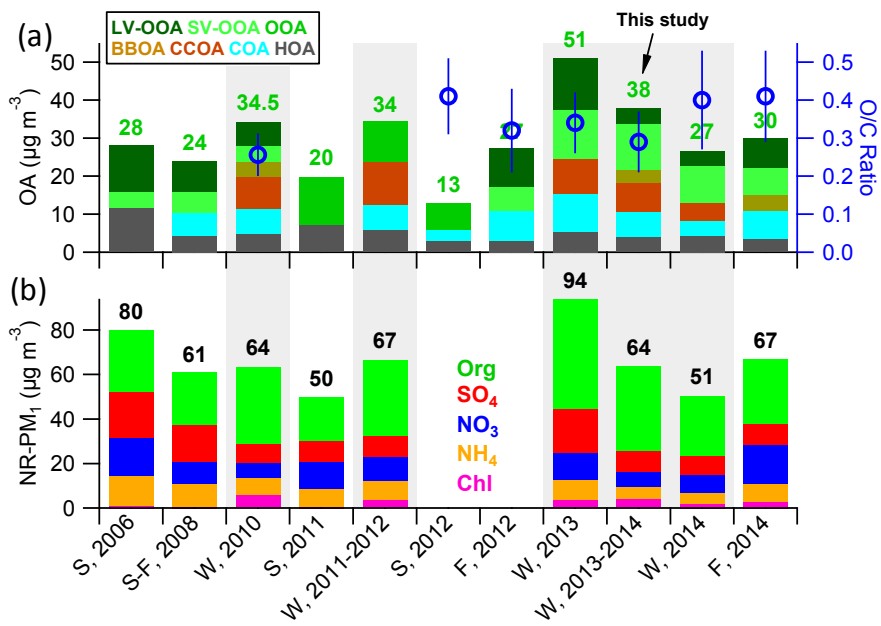

**Figure 2.** Average chemical composition of (a) organic aerosols and (b) NR-PM$_1$ in the megacity of Beijing measured by aerosol mass spectrometers. Also shown in (a) is the oxygen-to-carbon (O/C) ratio of organic aerosol for each study. The O/C was calculated using A-A method (Aiken et al., 2008). A more detailed description of the data is presented in Table S1.

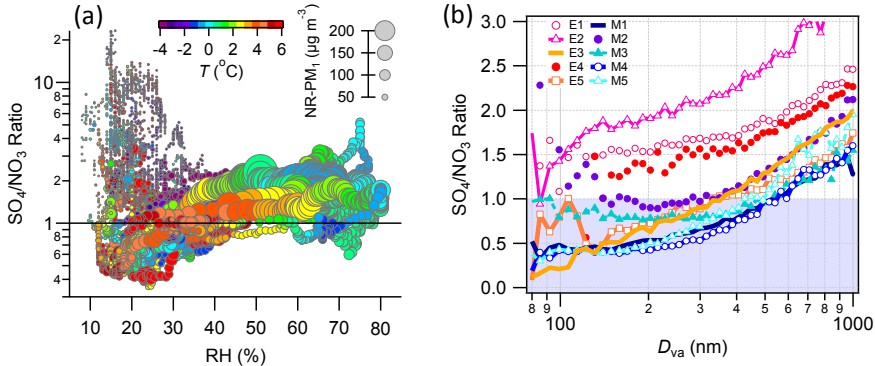

**Figure 3.** (a) Variations of SO$_4$/NO$_3$ ratios as a function of RH, (b) size-resolved SO$_4$/NO$_3$ ratio during five episodes (E1 – E5) with high RH levels (> 40%) and five episodes (M1 – M5) with low RH levels (< 40%).





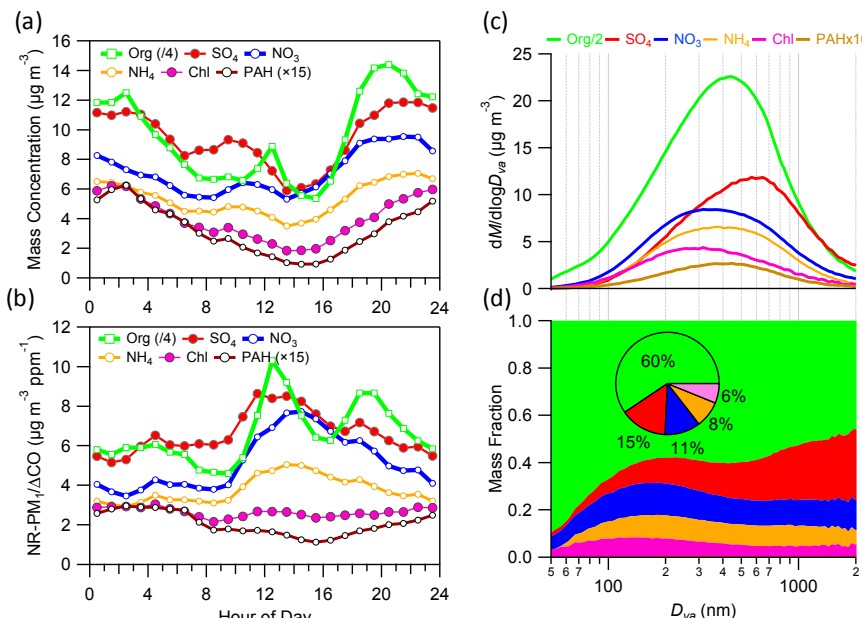

**Figure 4.** Average diurnal cycles of (a) NR-PM$_1$ species and (b) NR-PM$_1$ species/$\Delta$CO, (c) and (d) show the average mass size distributions of NR-PM$_1$ species. The pie chart in (d) shows the average chemical composition of NR-PM$_1$ for the entire study period.





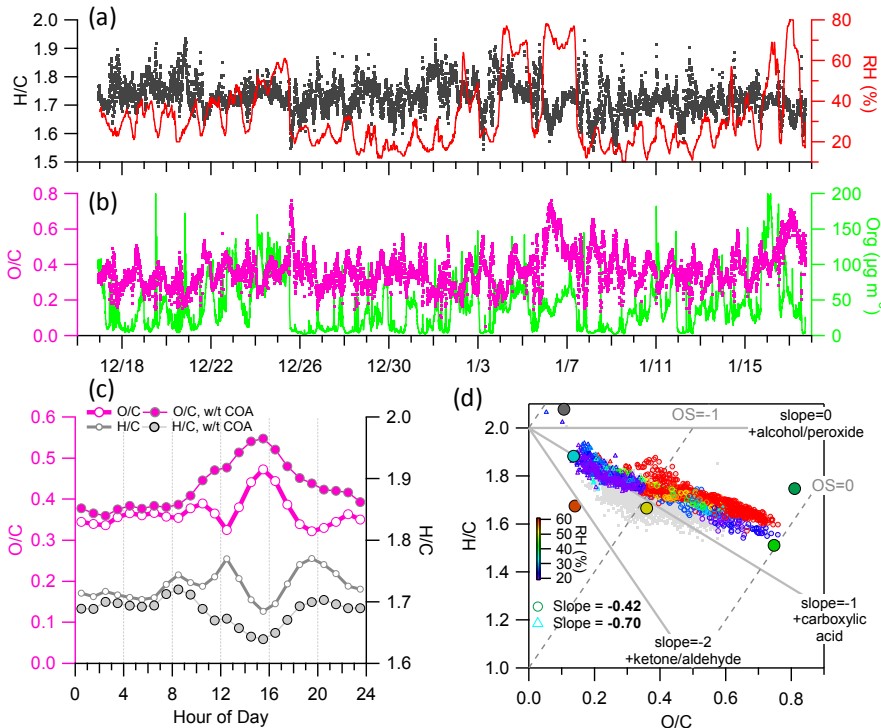

**Figure 5.** Time series of (a) H/C and (b) O/C ratios; (c) Average diurnal cycles of O/C and H/C. Also shown are the average diurnal cycles of elemental ratios by excluding the contribution of cooking organics aerosol, and (d) Van Krevelen diagram of H/C versus O/C. The RH color-coded triangle and circle points represent the data with the contributions of LV-OOA and COA being larger than 20% and 40%, respectively.



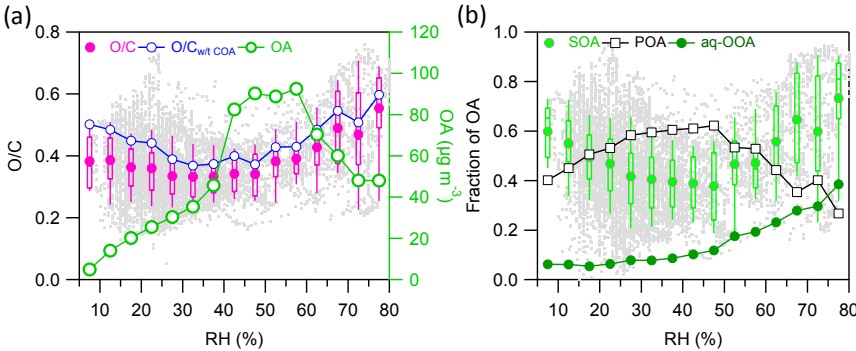

**Figure 6.** Variations of (a) O/C and OA, (b) mass fractions of POA, SOA, and LV-OOA as a function of RH. The data points are grouped in RH bins (5% increment).

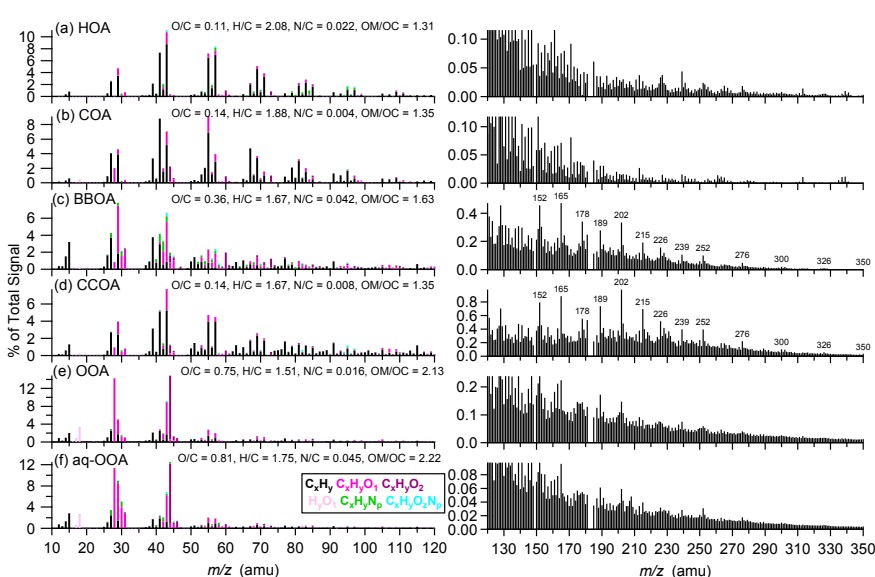

**Figure 7.** left panel: high resolution mass spectra of six OA factors; right panel: unit mass resolution spectra (*m/z* 120 – 350) of six OA factors.





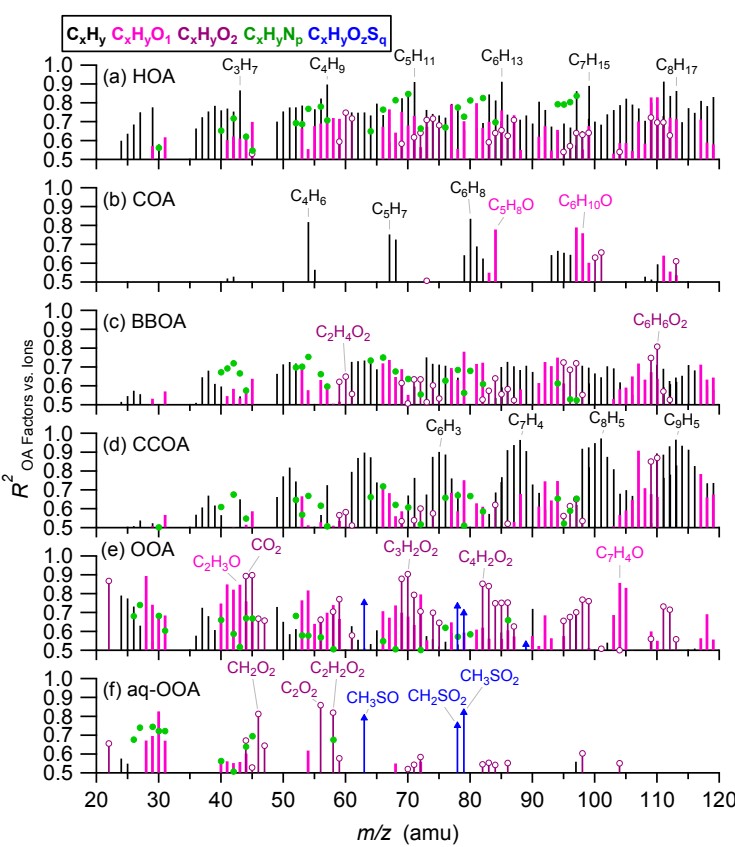

**Figure 8.** Correlations between six OA factors and HRMS ions that are segregated into five categories ($C_xH_y^+$, $C_xH_yO^+$, $C_xH_yO_2^+$, $C_xH_yN_p^+$, and $C_xH_yO_zS_q^+$).



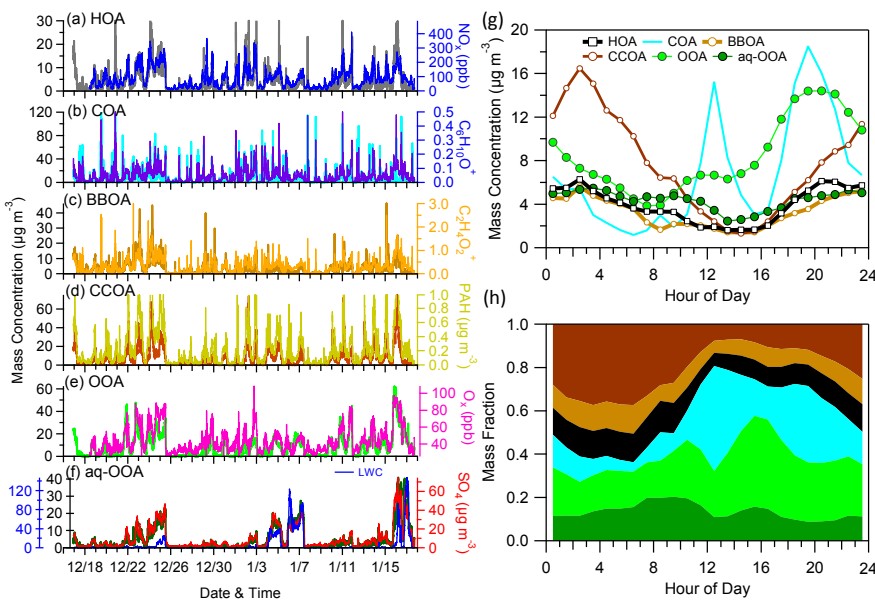

**Figure 9.** Time series of six OA factors and average diurnal cycles of mass concentrations and mass fractions of OA factors. The external tracer species including $NO_x$, $C_6H_{10}O^+$, $C_2H_4O_2^+$, PAH, $O_x$, and $SO_4$ are also shown for comparisons.

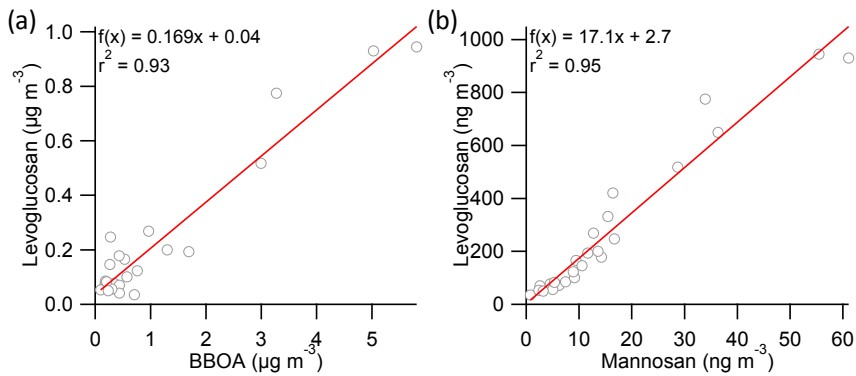

**Figure 10.** Correlations between levoglucosan and (a) BBOA, and (b) Mannosan.




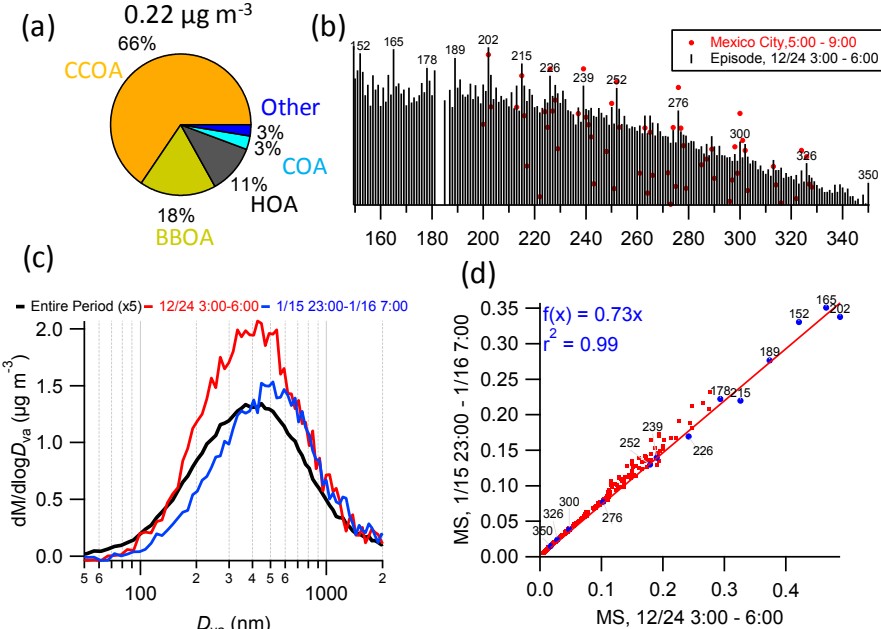

**Figure 11.** (a) Average contributions of OA factors to PAHs for the entire study, (b) UMR OA spectra during an episode (3:00 – 6:00 on December 24). The average PAH spectrum between 5:00 - 9:00 hours in Mexico City (Dzepina et al., 2007) is shown for a comparison. (c) shows the average size distributions of PAHs for the entire study and two selected episodes, and (d) presents a comparison of OA spectra (m/z 150 – 300) between the two episodes.





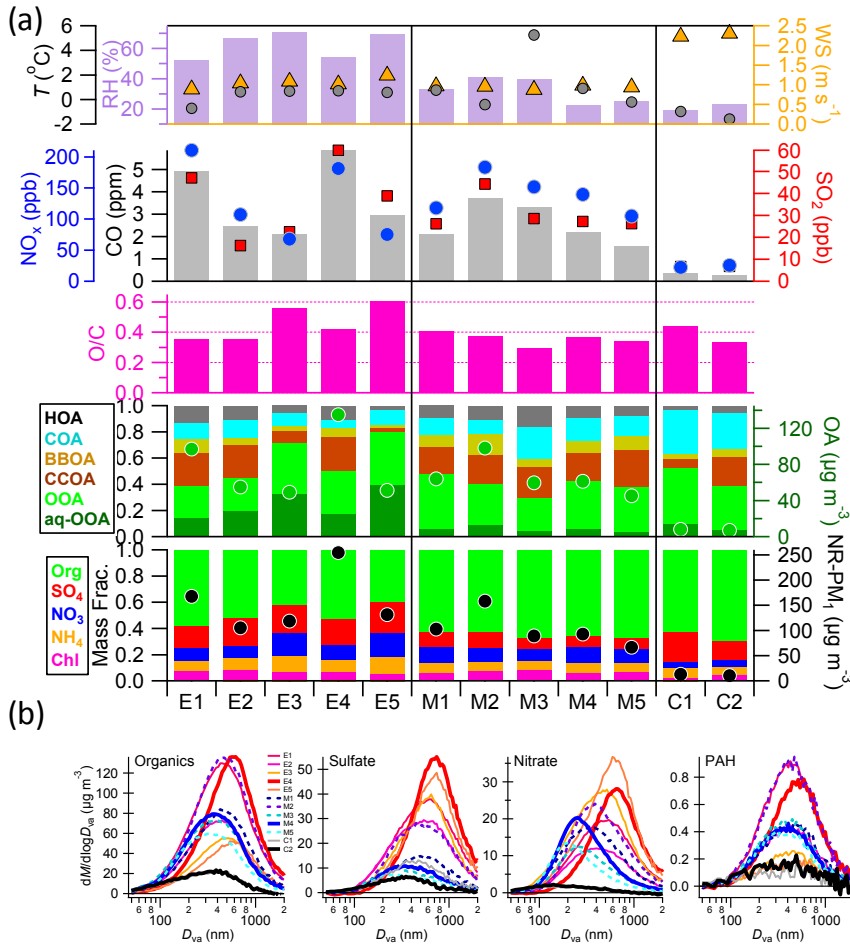

**Figure 12.** Summary of (a) meteorological conditions, gaseous species, O/C, OA composition and NR-PM$_1$ composition, and (b) size distributions of organics, sulfate, nitrate and chloride for episodes (E1 – E5, M1 – M5, C1 – C2) in Fig. 1.



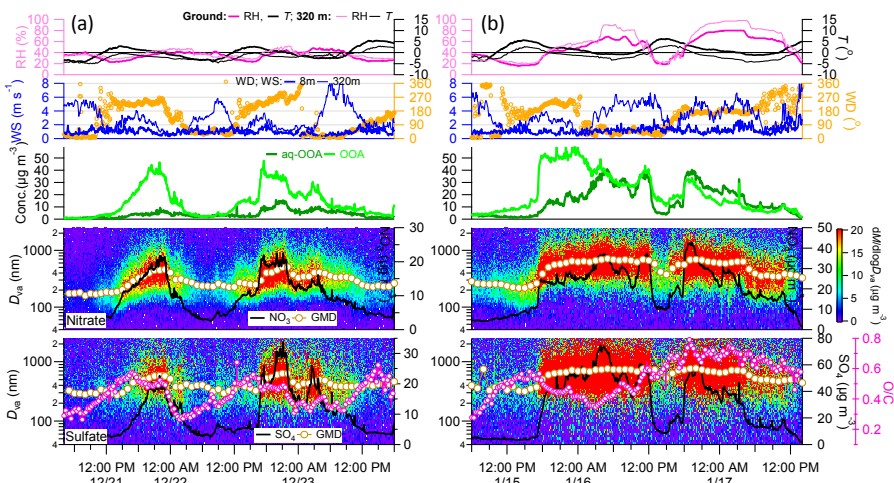

**Figure 13.** Evolution of meteorological parameters, SOA factors, O/C ratios, and size distributions of sulfate and nitrate during two different types of episodes. The O/C ratios in the figures were calculated by excluding the contributions of COA.

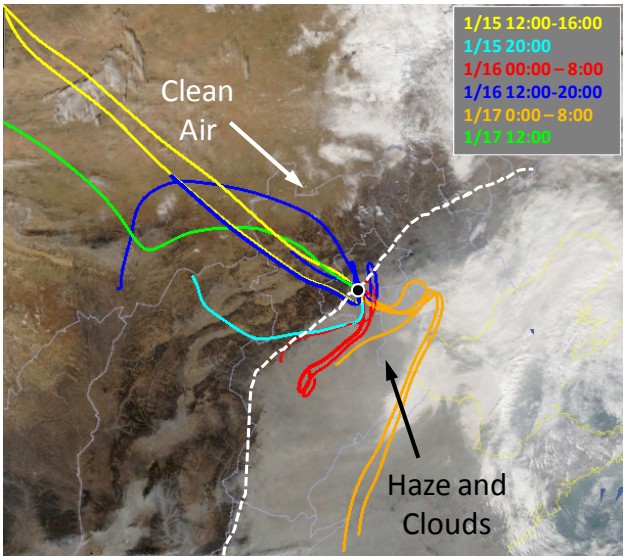

**Figure 14.** Two day back trajectories arriving at IAP, Beijing between 12:00 on January 15 and 12:00 on January 17. The back trajectories at 100 m height were calculated every 4 h using NOAA HYSPLIT Trajectory Model (http://www.ready.noaa.gov) (Stein et al., 2015).The background picture the MODIS image on January 16. The white dash line is an approximate dividing line between clean

10    and haze regions.