# Peer review of "Primary and secondary aerosols in Beijing in winter: sources, variations and processes"

_Atmospheric Chemistry and Physics, 2016_

## Referee Comment (RC1) · Anonymous Referee #1 · 15 May 2016

**Referee comment on**

**"Primary and secondary aerosols in Beijing in winter: sources, variations and processes"**

**by Sun et al., Atmos. Chem. Phys. Discuss., doi:10.5194/acp-2016-255, 2016**

**Anonymous referee #1**

**1.  General comments**

This manuscript reports results obtained during a field campaign performed at Beijing in Winter 2013/14. The authors deployed an Aerodyne high-resolution time-of-flight aerosol mass spectrometer (HR-ToF-AMS) to measure the particle concentration, chemical composition and size distribution, sampled particles on filters for subsequent extraction and analysis by gas chromatography/mass spectrometry (GC/MS), and measured gaseous species and meteorological data. A source apportionment of organics was performed by positive matrix factorization. The effect of the relative humidity on the particle concentration and chemical composition was studied through several case studies.

This manuscript is very descriptive, but well written and interesting. Moreover, with the severe pollution events occurring regularly at Beijing, it is very important to perform this kind of study in order to better understand sources and processes of particles impacting this megacity. Thus, I recommend its publication after the authors address the following comments.

**2.  Specific comments**

Page 4, line 9: I think that the reference Canagaratna et al. (2007) is much more appropriate here than Canagaratna et al. (2015).

Section 2 "Experimental methods": the authors give later in the manuscript some results from back trajectory analysis with the HYSPLIT model. They should describe this analysis in the "Experimental methods" section rather than in the caption of Figure 14. By the way, the back trajectory analysis reported here concerns only a short period (Jan 15th-17th). The absence of a complete analysis for the entire study is maybe the main weakness of this manuscript.

Section 2 "Experimental methods": it seems that all the dates and time are given in local time. The authors should mention that somewhere in this section.

Page 6, lines 26-27: a constant collection efficiency of 0.5 was used for this dataset. The authors need to justify this choice in the manuscript, in particular by giving some information on the particle acidity and on the presence or absence of a dryer in front of the AMS. Concerning the chemical composition, Figure 1f suggests that particles were never dominated by ammonium nitrate, so this point should be mentioned as well.

Section 3.1 "Mass concentrations and compositions": there is a long discussion on the $SO_4/NO_3$ ratio, without any information under which form these two species are present. So here also, a few words on the particle acidity would be helpful to clarify this point.

Page 10, lines 20-21: the authors claim that cooking organic aerosols (COA) are mainly in the ultrafine range (< 100 nm). This is in contradiction with results obtained by Ge et al. (2012), who showed that their COA factor had a very broad size distribution peaking at 450 nm (in $D_{va}$). However, according to other studies, the sizes of cooking-related particles vary widely, depending on cooking types, operations, and distance from the cooking sources. The authors may include this discussion in the manuscript.

Page 10, lines 26-28: the fact that two species have similar size distributions does not necessarily mean that they are internally mixed. This kind of information cannot be obtained with the AMS, which does not perform single particle analysis (unless the instrument is equipped with a light scattering module).

Page 14, line 11: concerning the Paris megacity, the authors can also mention the more recent study performed by Fröhlich et al. (2015), who also found a higher contribution of COA (15.0% of the total organics) than HOA (14.3%).

**3. Technical corrections**

Page 3, line 3: "is of a-great concern".

Page 3, line 5: "concentration of $PM_{2.5}$ in Beijing was decreased from".

Page 4, line 15: please define the "SIA" abbreviation.

Page 7, line 4: "hydrogen-to-carbon (H/C), nitrogen-to-carbon_(N/C), and".

Page 16, line 21: "levoglucosan /O/C of 0.062".

Page 17, line 10: "OOA was highly correlated to_the oxygenated ions series".

Page 19, line 4: "are of a-great concern".

Page 21, lines 4-5: "two episodes, i.e., E3 and E5 showed much higher SOA contributions (67 – 77%) than the other episodes".

Page 21, line 8: "enhanced the oxidation stageslevels of OA".

Page 40, Figure 12b: this figure is very crowded, the different size distributions are a bit hard to identify. In particular, the colors of E1, E2 and E4 are almost similar, as well as those of M3 and M5. The authors may try to use different markers, it will be easier to identify the different size distributions.

Page 41, caption of Figure 14: "(Stein et al., 2015)._The background picture is_the MODIS image on January 16". This sentence is incomplete, I'm not sure whether that's what the authors wanted to say.

**4. References**

Canagaratna, M. R., Jayne, J. T., Jimenez, J. L., Allan, J. D., Alfarra, M. R., Zhang, Q., Onasch, T. B., Drewnick, F., Coe, H., Middlebrook, A., Delia, A., Williams, L. R., Trimborn, A. M., Northway, M. J., DeCarlo, P. F., Kolb, C. E., Davidovits, P., and Worsnop, D. R.: Chemical and microphysical characterization of ambient aerosols with the Aerodyne aerosol mass spectrometer, Mass Spectrom. Rev., 26, 185-222, 10.1002/mas.20115, 2007.

Canagaratna, M. R., Jimenez, J. L., Kroll, J. H., Chen, Q., Kessler, S. H., Massoli, P., Hildebrandt Ruiz, L., Fortner, E., Williams, L. R., Wilson, K. R., Surratt, J. D., Donahue, N. M., Jayne, J. T., and Worsnop, D. R.: Elemental ratio measurements of organic compounds using aerosol mass spectrometry: characterization, improved calibration, and implications, Atmos. Chem. Phys., 15, 253-272, 10.5194/acp-15-253-2015, 2015.

Fröhlich, R., Crenn, V., Setyan, A., Belis, C. A., Canonaco, F., Favez, O., Riffault, V., Slowik, J. G., Aas, W., Aijälä, M., Alastuey, A., Artiñano, B., Bonnaire, N., Bozzetti, C., Bressi, M., Carbone, C., Coz, E., Croteau, P. L., Cubison, M. J., Esser-Gietl, J. K., Green, D. C., Gros, V., Heikkinen, L., Herrmann, H., Jayne, J. T., Lunder, C. R., Minguillón, M. C., Močnik, G., O'Dowd, C. D., Ovadnevaite, J., Petralia, E., Poulain, L., Priestman, M., Ripoll, A., Sarda-Estève, R., Wiedensohler, A., Baltensperger, U., Sciare, J., and Prévôt, A. S. H.: ACTRIS ACSM intercomparison – Part 2: Intercomparison of ME-2 organic source apportionment results from 15 individual, co-located aerosol mass spectrometers, Atmos. Meas. Tech., 8, 2555-2576, 10.5194/amt-8-2555-2015, 2015.

Ge, X., Setyan, A., Sun, Y., and Zhang, Q.: Primary and secondary organic aerosols in Fresno, California during wintertime: Results from high resolution aerosol mass spectrometry, J. Geophys. Res., 117, D19301, 10.1029/2012jd018026, 2012.

---

## Referee Comment (RC2) · Anonymous Referee #2 · 17 Jun 2016

This manuscript presents a comprehensive study during wintertime in Beijing using a suit of instruments. The data analysis is very thoughtful and focusing on the aqueous processing of secondary aerosol on the polluted urban area during wintertime. In addition, this work present substantial analysis on PMF analysis and show a convince PMF result. Overall, this is well written and organized. I recommend it is publication on ACP after a minor revision following the below comments. 1. One of major finding in this manuscript is aqueous processing on particulate water. However, the authors use relative humidity to demonstrate the aqueous chemistry which is at the function of atmospheric pressure and temperature. I suggest the author using specific humidity other than relative humidity. 2. The mass spectrum of CCOA of this work is characterized by the high signal at m/z 115 which is similar with previous studies. However, there are also significant signal at m/z 44 from CCOA in previous studies, such as Beijing (Hu

et al., 2016), Lanzhou (Xu et al., 2016), and laboratory study (Dall'Osto et al., 2016) which could from the organic acids during coal burning. Please add the comparison and give the explanation for these differences. 3. It seems that the source of BBOA in this study from both the biomass burning and soft coal combustion based on the L/M ratio. These emissions may not from the residential or industry in Beijing due to the strict control, and could from the regional transport. So present the bivariate polar plots in supporting information could show some evidences on the source of BBOA. 4. P4, line13-14: please cite the reference at the order of year. 5. P6, line4: 350 nm ammonium nitrate is obtained from SMPS or AMS? 6. P6, line 25: please present the full name of NR-PM1. 7. P7, line 21-22: please show the R2 .

---

## Author Comment (AC1) · 23 Jun 2016

**Response to Reviewer's comments**

We are thankful to the two reviewers for their thoughtful and constructive comments that help improve the manuscript substantially. We have revised the manuscript accordingly. Listed below is our point-to-point response in blue to each comment that was offered by the reviewers.

**Response to Reviewer #1**

**1. General comments**

This manuscript reports results obtained during a field campaign performed at Beijing in Winter 2013/14. The authors deployed an Aerodyne high-resolution time-of-flight aerosol mass spectrometer (HR-ToF-AMS) to measure the particle concentration, chemical composition and size distribution, sampled particles on filters for subsequent extraction and analysis by gas chromatography/mass spectrometry (GC/MS), and measured gaseous species and meteorological data. A source apportionment of organics was performed by positive matrix factorization. The effect of the relative humidity on the particle concentration and chemical composition was studied through several case studies.

This manuscript is very descriptive, but well written and interesting. Moreover, with the severe pollution events occurring regularly at Beijing, it is very important to perform this kind of study in order to better understand sources and processes of particles impacting this megacity. Thus, I recommend its publication after the authors address the following comments.

We thank the reviewer for his/her positive comments on this manuscript.

**2. Specific comments**

Page 4, line 9: I think that the reference Canagaratna et al. (2007) is much more appropriate here than Canagaratna et al. (2015).

Right, Canagaratna et al. (2007) was cited.

Section 2 "Experimental methods": the authors give later in the manuscript some results from back trajectory analysis with the HYSPLIT model. They should describe this analysis in the "Experimental methods" section rather than in the caption of Figure 14. By the way, the back trajectory analysis reported here concerns only a short period (Jan 15th-17th). The absence of a complete analysis for the entire study is maybe the main weakness of this manuscript.

We thank the reviewer's comments. The back trajectory analysis has been widely used to investigate the sources of air masses in Beijing in previous AMS studies (Huang et al., 2010; Sun et al., 2010; Sun et al., 2012a; Zhang et al., 2015), and it has been well recognized that

clean periods with low mass loadings are typically associated with air masses from the north and northwest while polluted episodes are often primarily related to the southern and southwestern air masses. Therefore, we feel that it is not necessary to do such a similar analysis in this study. The back trajectory analysis in Fig. 14 was mainly used to support the discussions on the evolution of E4 that was tightly associated with air mass changes and also the spatial distribution of haze. Because the HYSPLIT model has been well developed and the descriptions have been detailed in many previous studies, only a brief description of the trajectory height and time resolution was given in the caption of Fig. 14.

Section 2 "Experimental methods": it seems that all the dates and time are given in local time. The authors should mention that somewhere in this section.
The sentence "All the data in this study are reported at ambient temperature and pressure conditions in Beijing Standard Time (BST), which equals Coordinated Universal Time (UTC) plus 8 h." was added in Section 2 in the revised manuscript.

Page 6, lines 26-27: a constant collection efficiency of 0.5 was used for this dataset. The authors need to justify this choice in the manuscript, in particular by giving some information on the particle acidity and on the presence or absence of a dryer in front of the AMS. Concerning the chemical composition, Figure 1f suggests that particles were never dominated by ammonium nitrate, so this point should be mentioned as well.
We thank the reviewer for pointing this out. The reasons for the selection of CE = 0.5 were expanded in the revised manuscript. It now reads: "A collection efficiency (CE) of 0.5 was applied to the entire dataset to compensate for the incomplete detection of the AMS because (1) aerosol particles were dried by a silica gel dryer, (2) aerosol particles were slightly acidic (Fig. S1), yet not high enough to affect CE substantially, and (3) the mass fraction of $NH_4NO_3$ was smaller than 0.4 for the entire study (Matthew et al., 2008; Middlebrook et al., 2012). "

[Figure]

Figure S1. Correlation between measured $NH_4^+$ and predicted $NH_4^+$ (=18× (2×$SO_4^{2-}$/96 + $NO_3^-$/62 + Chl/35.5)).

Section 3.1 "Mass concentrations and compositions": there is a long discussion on the SO4/NO3 ratio, without any information under which form these two species are present. So here also, a few words on the particle acidity would be helpful to clarify this point.

Good point. We added the following sentence in the text.

"Considering that aerosol particles were slightly acidic as indicated by the average ratio (0.69) of measured $NH_4^+$ to predicted $NH_4^+$ that requires to fully neutralize sulfate, nitrate and chloride (Zhang et al., 2007), sulfate in this study mainly existed in the form of $(NH_4)_2SO_4$ and $NH_4HSO_4$."

Page 10, lines 20-21: the authors claim that cooking organic aerosols (COA) are mainly in the ultrafine range (< 100 nm). This is in contradiction with results obtained by Ge et al. (2012), who showed that their COA factor had a very broad size distribution peaking at 450 nm (in D$va$). However, according to other studies, the sizes of cooking-related particles vary widely, depending on cooking types, operations, and distance from the cooking sources. The authors may include this discussion in the manuscript.

We agree with the reviewer that COA factor generally has a very broad size distribution, e.g., in Fresno (Ge et al., 2012), New York City (Sun et al., 2012b), and Lanzhou (Xu et al., 2014). The sentence here was not meant to discuss the size distribution of COA, instead, it emphasized that the ultrafine particles were mainly contributed by COA. COA can have a broad size distribution with higher concentration at larger size, but its contribution to ultrafine particles was larger than other OA factors. We are sorry that this statement was misunderstood.

Page 10, lines 26-28: the fact that two species have similar size distributions does not necessarily mean that they are internally mixed. This kind of information cannot be obtained with the AMS, which does not perform single particle analysis (unless the instrument is equipped with a light scattering module).

We agree with the reviewer and revised this sentence as:

"While the two species shared the similar size distributions in summer, they peaked at different sizes which are ~600 nm and ~300 – 400 nm, respectively."

Page 14, line 11: concerning the Paris megacity, the authors can also mention the more recent study performed by Fröhlich et al. (2015), who also found a higher contribution of COA (15.0% of the total organics) than HOA (14.3%).

Good point. The reference Fröhlich et al. (2015) was cited in the revised manuscript.

**3. Technical corrections**

Page 3, line 3: "is of a great concern".

Page 3, line 5: "concentration of PM2.5 in Beijing was decreased from".

Corrected.

Page 4, line 15: please define the "SIA" abbreviation.

It was defined as "SOA and secondary inorganic aerosols (SIA)".

Page 7, line 4: "hydrogen-to-carbon (H/C), nitrogen-to-carbon_(N/C), and".

Page 16, line 21: "levoglucosan /O/C of 0.062".

Page 17, line 10: "OOA was highly correlated to_the oxygenated ions series".

Page 19, line 4: "are of a great concern".

Page 21, lines 4-5: "two episodes, i.e., E3 and E5 showed much higher SOA contributions (67 – 77%) than the other episodes".

Page 21, line 8: "enhanced the oxidation levels of OA".

The typos and grammar mistakes were corrected in the revised manuscript.

Page 40, Figure 12b: this figure is very crowded, the different size distributions are a bit hard to identify. In particular, the colors of E1, E2 and E4 are almost similar, as well as those of M3 and M5. The authors may try to use different markers, it will be easier to identify the different size distributions.

This figure was revised for easy reading (see below).

[Figure]

**Figure 12.** Summary of (a) meteorological conditions, gaseous species, O/C, OA composition and NR-PM$_1$ composition, and size distributions of organics, sulfate, nitrate and chloride for episodes (b) E1 – E5 and (c) M1 – M5 and C1 – C2 in Fig. 1.

Page 41, caption of Figure 14: "(Stein et al., 2015)._The background picture is_the MODIS image on January 16". This sentence is incomplete, I'm not sure whether that's what the authors wanted to say.

The background picture shows the image from the Moderate Resolution Imaging Spectroradiometer (MODIS) instrument on board the Aqua Satellite on January 16. Such descriptions were added in the caption.

**Response to Reviewer #2**

This manuscript presents a comprehensive study during wintertime in Beijing using a suit of instruments. The data analysis is very thoughtful and focusing on the aqueous processing of secondary aerosol on the polluted urban area during wintertime. In addition, this work present substantial analysis on PMF analysis and show a convince PMF result. Overall, this is well written and organized. I recommend it is publication on ACP after a minor revision following the below comments.

1.      One of major finding in this manuscript is aqueous processing on particulate water. However, the authors use relative humidity to demonstrate the aqueous chemistry which is at the function of atmospheric pressure and temperature. I suggest the author using specific humidity other than relative humidity.

We thank the reviewer's comments. We used relative humidity rather than specific humidity mainly due to two reasons: (1) to be consistent with many previous studies for studying aqueous-phase processing (Hennigan et al., 2008; Hennigan et al., 2009; Zhang et al., 2012); (2) relative humidity was highly related to aerosol liquid water (Figure R1) that was estimated by ISORROPIA-II model. Because we didn't have the direct measurements of aerosol liquid water content, relative humidity was used in this study. We agree with the reviewer that specific humidity by considering atmospheric pressure and temperature could be another choice, which can be explored in future studies.

[Figure]

Figure R1. Relationship between liquid water content and relative humidity (RH).

2.      The mass spectrum of CCOA of this work is characterized by the high signal at m/z 115 which is similar with previous studies. However, there are also significant signal at m/z 44 from CCOA in previous studies, such as Beijing (Hu et al., 2016), Lanzhou (Xu et al., 2016), and laboratory study (Dall'Osto et al., 2016) which could from the organic acids during coal burning. Please add the comparison and give the explanation for these differences.

The discussions on spectral differences were expanded in the revised manuscript.

"The CCOA spectrum showed a visible *m/z* 44 peak, yet it is much smaller than those observed in Beijing (Hu et al., 2016) and Lanzhou (Xu et al., 2016). The spectral differences were likely due to different burning conditions and ageing processes. For example, a recent study by burning different types of coals showed that the CCOA spectrum, particularly *m/z* 44 and 73, can have significant changes during the different stages of the burning, and high *m/z* 44 in the spectrum was mainly from fragmentation of organic acids (Zhou et al., 2016)."

3.      It seems that the source of BBOA in this study from both the biomass burning and soft coal combustion based on the L/M ratio. These emissions may not from the residential or industry in Beijing due to the strict control, and could from the regional transport. So present the bivariate polar plots in supporting information could show some evidences on the source of BBOA.

Thank the reviewer's comments. As indicated by the bivariate polar plot below, high concentration of BBOA was mainly located in the north and northeast regions (WS < 15 km hr$^{-1}$). This indicates that BBOA in this study was likely from a mix of local sources and regional transport. Although strict controls on the use of coal were implemented inside the city, i.e., 5[th] ring road, biomass burning and coal combustion were still significant outside the city, particularly in rural areas in Beijing. In addition, sporadic residential coal combustions were often observed inside the city. Therefore, it seems to be difficult to distinguish local sources from regional transport based on current available data. Following the reviewer's comments, the bivariate polar plot of BBOA was added in the revised manuscript.

[Figure]

Figure R2. Bivariate polar plot of BBOA.

4. P4, line13-14: please cite the reference at the order of year.

The in-text citations were now chronologically listed.

5. P6, line4: 350 nm ammonium nitrate is obtained from SMPS or AMS?

350 nm is mobility diameter from the SMPS measurement, which was clarified in the revised manuscript.

6. P6, line 25: please present the full name of NR-PM1.

NR-PM$_1$, i.e., non-refractory submicron aerosol, was spelled out in the revised manuscript.

7. P7, line 21-22: please show the R2.

$R^2$ was added.

References

Fröhlich, R., Crenn, V., Setyan, A., Belis, C. A., Canonaco, F., Favez, O., Riffault, V., Slowik, J. G., Aas, W., Aijälä, M., Alastuey, A., Artiñano, B., Bonnaire, N., Bozzetti, C., Bressi, M., Carbone, C., Coz, E., Croteau, P. L., Cubison, M. J., Esser-Gietl, J. K., Green, D. C., Gros, V., Heikkinen, L., Herrmann, H., Jayne, J. T., Lunder, C. R., Minguillón, M. C., Močnik, G., O'Dowd, C. D., Ovadnevaite, J., Petralia, E., Poulain, L., Priestman, M., Ripoll, A., Sarda-Estève, R., Wiedensohler, A., Baltensperger, U., Sciare, J., and Prévôt, A. S. H.: ACTRIS ACSM intercomparison – Part 2: Intercomparison of ME-2 organic source apportionment results from 15 individual, co-located aerosol mass spectrometers, Atmos. Meas. Tech., 8, 2555-2576, 10.5194/amt-8-2555-2015, 2015.

Ge, X., Setyan, A., Sun, Y., and Zhang, Q.: Primary and secondary organic aerosols in Fresno, California during wintertime: Results from high resolution aerosol mass spectrometry, J. Geophys. Res., 117, D19301, 10.1029/2012JD018026, 2012.

Hennigan, C. J., Bergin, M. H., Dibb, J. E., and Weber, R. J.: Enhanced secondary organic aerosol formation due to water uptake by fine particles, Geophys. Res. Lett., 35, L18801, doi:18810.11029/12008GL035046, 2008.

Hennigan, C. J., Bergin, M. H., Russell, A. G., Nenes, A., and Weber, R. J.: Gas/particle partitioning of water-soluble organic aerosol in Atlanta, Atmos. Chem. Phys., 9, 3613-3628, 2009.

Hu, W., Hu, M., Hu, W., Jimenez, J. L., Yuan, B., Chen, W., Wang, M., Wu, Y., Chen, C., Wang, Z., Peng, J., Zeng, L., and Shao, M.: Chemical composition, sources and aging process of sub-micron aerosols in Beijing: contrast between summer and winter, J. Geophys. Res., 121, 1955-1977, 10.1002/2015JD024020, 2016.

Huang, X. F., He, L. Y., Hu, M., Canagaratna, M. R., Sun, Y., Zhang, Q., Zhu, T., Xue, L., Zeng, L. W., Liu, X. G., Zhang, Y. H., Jayne, J. T., Ng, N. L., and Worsnop, D. R.: Highly time-resolved chemical characterization of atmospheric submicron particles during 2008 Beijing Olympic Games using an Aerodyne High-Resolution Aerosol Mass Spectrometer, Atmos. Chem. Phys., 10, 8933-8945, 10.5194/acp-10-8933-2010, 2010.

Matthew, B. M., Middlebrook, A. M., and Onasch, T. B.: Collection efficiencies in an Aerodyne Aerosol Mass Spectrometer as a function of particle phase for paboratory generated aerosols, Aerosol Sci. Tech., 42, 884 - 898, 2008.

Middlebrook, A. M., Bahreini, R., Jimenez, J. L., and Canagaratna, M. R.: Evaluation of composition-dependent collection efficiencies for the Aerodyne Aerosol Mass Spectrometer using field data, Aerosol Sci. Tech., 46, 258-271, 2012.

Sun, J., Zhang, Q., Canagaratna, M. R., Zhang, Y., Ng, N. L., Sun, Y., Jayne, J. T., Zhang, X., Zhang, X., and Worsnop, D. R.: Highly time- and size-resolved characterization of submicron aerosol particles in Beijing using an Aerodyne Aerosol Mass Spectrometer, Atmos. Environ., 44, 131-140, 2010.

Sun, Y. L., Wang, Z., Dong, H., Yang, T., Li, J., Pan, X., Chen, P., and Jayne, J. T.: Characterization of summer organic and inorganic aerosols in Beijing, China with an Aerosol Chemical Speciation Monitor, Atmos. Environ., 51, 250-259, 10.1016/j.atmosenv.2012.01.013, 2012a.

Sun, Y. L., Zhang, Q., Schwab, J. J., Yang, T., Ng, N. L., and Demerjian, K. L.: Factor analysis of combined organic and inorganic aerosol mass spectra from high resolution aerosol mass spectrometer measurements, Atmos. Chem. Phys., 12, 8537-8551, 10.5194/acp-12-8537-2012, 2012b.

Xu, J., Zhang, Q., Chen, M., Ge, X., Ren, J., and Qin, D.: Chemical composition, sources, and processes of urban aerosols during summertime in northwest China: insights from high-resolution aerosol mass spectrometry, Atmos. Chem. Phys., 14, 12593-12611, 10.5194/acp-14-12593-2014, 2014.

Xu, J., Shi, J., Zhang, Q., Ge, X., Canonaco, F., Prévôt, A. S. H., Vonwiller, M., Szidat, S., Ge, J., Ma, J., An, Y., Kang, S., and Qin, D.: Wintertime organic and inorganic aerosols in Lanzhou, China: Sources, processes and comparison with the results during summer, Atmos. Chem. Phys. Discuss., 2016, 1-52, 10.5194/acp-2016-278, 2016.

Zhang, J., Wang, Y., Huang, X., Liu, Z., Ji, D., and Sun, Y.: Characterization of organic aerosols in Beijing using an aerodyne high-resolution aerosol mass spectrometer, Advances in Atmospheric Sciences, 32, 877-888, 10.1007/s00376-014-4153-9, 2015.

Zhang, Q., Jimenez, J. L., Worsnop, D. R., and Canagaratna, M.: A case study of urban particle acidity and its effect on secondary organic aerosol, Environ. Sci. Technol., 41, 3213-3219, 2007.

Zhang, X., Liu, J., Parker, E. T., Hayes, P. L., Jimenez, J. L., de Gouw, J. A., Flynn, J. H., Grossberg, N., Lefer, B. L., and Weber, R. J.: On the gas-particle partitioning of soluble organic aerosol in two urban atmospheres with contrasting emissions: 1. Bulk water-soluble organic carbon, J. Geophys. Res., 117, D00V16, 10.1029/2012jd017908, 2012.

Zhou, W., Jiang, J., Duan, L., and Hao, J.: Evolution of submicron organic aerosols during a complete residential coal combustion process, Environ. Sci. Technol., 10.1021/acs.est.6b00075, 2016.